# ADAPT-AND-ADJUST: OVERCOMING THE LONG-TAIL PROBLEM OF MULTILINGUAL SPEECH RECOGNITION

## ABSTRACT

One crucial challenge of real-world multilingual speech recognition is the long-tailed distribution problem, where some resource-rich languages like English have abundant training data, but a long tail of low-resource languages have varying amounts of limited training data. To overcome the long-tail problem, in this paper, we propose Adapt-and-Adjust (A2), a transformer-based multi-task learning framework for end-to-end multilingual speech recognition. The A2 framework overcomes the long-tail problem via three techniques: (1) exploiting a pretrained multilingual language model (mBERT) to improve the performance of low-resource languages; (2) proposing dual adapters consisting of both language-specific and language-agnostic adaptation with minimal additional parameters; and (3) overcoming the class imbalance, either by imposing class priors in the loss during training or adjusting the logits of the softmax output during inference. Extensive experiments on the CommonVoice corpus show that A2 significantly outperforms conventional approaches.

## 1 INTRODUCTION

Deploying a single Automatic Speech Recognition (ASR) model to recognize multiple languages is highly desired but very challenging for real-world multilingual ASR scenarios due to the well-known long-tailed distribution challenge, namely, that some resource-rich languages like English have abundant training data, while the majority low-resource languages have varying amounts of training data. The recent popular end-to-end (E2E) monolingual ASR architecture (Graves et al., 2013; Chan et al., 2015; Vaswani et al., 2017) is promising to achieve state-of-the-art performance for resource-rich languages but suffers dramatically from the long tail of low-resource languages due to the lack of training data. This paper aims to investigate an end-to-end multilingual ASR framework where a single model is trained end-to-end from a pooled dataset of all target languages to improve the overall performance of multilingual ASR tasks, especially for low-resource languages.

The long-tailed data distribution problem makes building an end-to-end multilingual ASR notoriously challenging. This imbalanced data setting poses a multitude of open challenges for multi-task training because the distribution of the training data is very skewed. These challenges stem from two aspects. First, very limited audio samples are available for low-resource languages, such as Kyrgyz, Swedish, and Turkish, while simultaneously, vast amounts of data exist from high-resource languages, such as English, French, and Spanish. Second, graphemes or subword labels follow a long-tailed distribution in ASR since some labels appear significantly more frequently, even for a monolingual setting. Furthermore, a multilingual system may include languages with writing scripts other than the Latin alphabet, such as Chinese or Cyrillic, that further worsen the skewness. To further illustrate the long-tail distribution in our study, Figure 1 shows the frequencies of sentence piece tokens in the curated multilingual dataset from CommonVoice (Ardila et al., 2020).

While a standard end-to-end multilingual training approach can improve overall performance compared with monolingual end-to-end approaches, it does not address the long-tail problem explicitly. One of the key challenges is the class imbalance issue, which will bias the multilingual model towards the dominant languages. To address this, one straightforward approach is to resample the training data (Kannan et al., 2019; Pratap et al., 2020) during batch assembly. However, such an ad-hoc approach does not fully resolve the underlying long-tail distribution problem, and only a marginal improvement is obtained in practice. Another challenge is how to model the languages

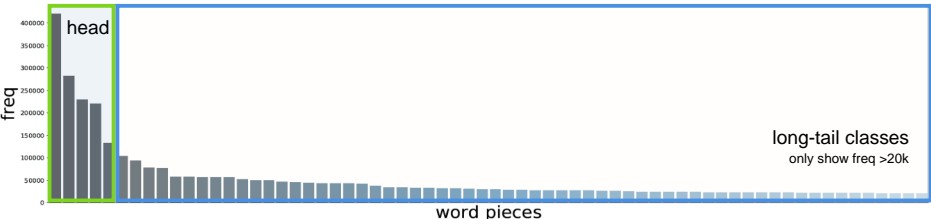

Figure 1: The long-tail distribution of sentence piece tokens in the curated multilingual dataset. The head classes are tokens with high frequency, otherwise, they are classified as tail classes.

with limited training data robustly. In this paper, the "long-tail problem" is twofold: 1) the long-tailed class distribution arising from the skewed multilingual data and sentence piece distribution 2) the robust modelling of languages with limited training data, *i.e.,* tail languages.

To this end, we propose the Adapt-and-Adjust (A2) framework for multilingual speech recognition using a speech transformer to address the twofold long-tail problem. Firstly, for better language modeling, a distilled mBERT (Devlin et al., 2019) is converted to an autoregressive transformer decoder to jointly explore the multilingual acoustic and text space to improve the performance of low-resource languages. Secondly, to adapt the multilingual network to specific languages with minimal additional parameters, both language-specific and language-agnostic adapters are used to augment each encoder and decoder layer. While the language-specific adapters focus on adapting the shared network weights to a particular language, a common adapter is proposed to learn some shared and language-agnostic knowledge for better knowledge transfer across languages. Lastly, to increase the relative margin between logits of rare versus dominant languages, we perform class imbalance adjustments during multilingual model training or inference by revisiting the classic idea of logit adjustment (Zhou & Liu, 2006). Class imbalance adjustment (Collell et al., 2016; Cui et al., 2019; Menon et al., 2020) is applied by adjusting the logits of the softmax input with the class priors. We conduct experiments and establish a benchmark from the CommonVoice corpus with a realistic long-tailed distribution of different languages. The extensive experiments show that A2 significantly outperforms conventional approaches for end-to-end multilingual ASR.

Our key contributions are as follows:

- We propose Adapt-and-Adjust (A2), a novel end-to-end transformer-based framework for real-world multilingual speech recognition to overcome the "long-tail problem";

- We demonstrate the effectiveness of utilizing a pretrained multilingual language model as a speech decoder to improve multilingual text representations and language adapters to better share the learned information across all languages. To the best of our knowledge, this work is the first to adapt a pretrained multilingual language model for multilingual ASR.

- We show that incorporating class priors during training or inference is effective and essential to addressing the long-tail distribution issue in multilingual training.

- We establish a reproducible multilingual speech recognition benchmark with long-tailed distributions of 11 languages from different language families for the research community.

## 2 ADAPT-AND-ADJUST FRAMEWORK

### 2.1 OVERVIEW

Figure 2 gives an overview of the proposed A2 framework for end-to-end multilingual ASR. A2 is built on a transformer-based sequence-to-sequence model with three key novel contributions: (1) an mBERT-based decoder, (2) language adapters, and (3) class-imbalance adjustments. Firstly, the vanilla transformer decoder is replaced with mBERT for better language modeling, particularly for low-resource languages. Secondly, the common and language-specific adapters are added to each encoder and decoder layer to learn both the shared and language-specific information for better acoustic modelling. Finally, we perform class imbalance adjustments during training or inference, where the logits are adjusted with the class priors estimated from the training data.

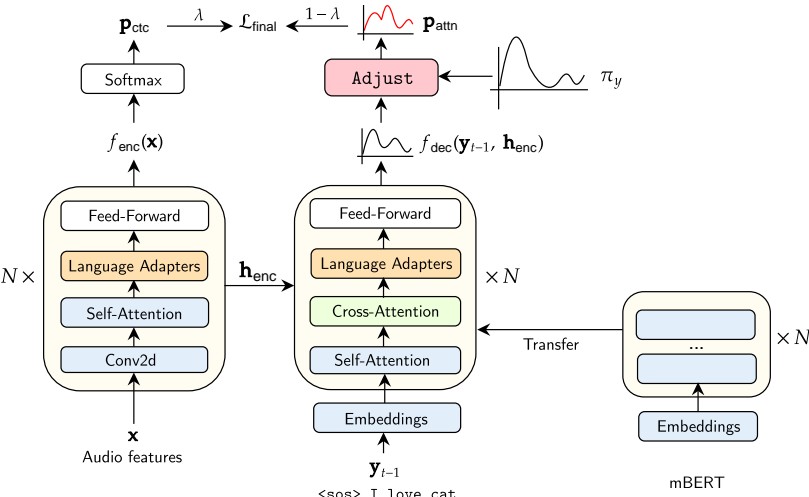

Figure 2: Overview of the Adapt-and-Adjust framework. The layer norm is omitted to save space. $\mathbf{p}_{ctc}$ is the connectionist temporal classification (CTC) output, $\mathbf{p}_{attn}$ is the decoder output, $\mathbf{y}_{t-1}$ is the previous token. The three key modules are (1) pre-trained mBERT-based decoding to improve language modelling; and (2) dual adapters to improve acoustic modelling, all particularly for tail languages; and (3) class-imbalance adjustment of the logits $f_{dec}(\mathbf{y}_{t-1}, \mathbf{h}_{enc})$ by class priors $\pi_y$.

## 2.2 BASE MODEL: HYBRID CTC-ATTENTION SPEECH TRANSFORMER

A sequence-to-sequence speech transformer model (Dong et al., 2018; Kim et al., 2016; Karita et al., 2019b) based on the hybrid CTC-Attention network is used for acoustic modeling. It takes in the acoustic features $\mathbf{x} \in \mathbb{R}^{T \times F}$ and outputs the sentence piece tokens $\mathbf{y}$, where $T$ and $F$ denote the sequence length and feature dimension. The encoder consists of several 2D convolution layers followed by self-attention layers. The convolution layers are used to extract more robust features before they are sent to the transformer. The decoder layers have two attention mechanisms, one for self-attention and the other for the encoder output. The network is trained in an autoregressive manner by predicting the next token given the current output. In addition, the CTC layer (Graves et al., 2006) is added to the encoder output to serve as a regularizer to the attention model.

**Training**  Multi-task loss $\mathcal{L}_{MTL}$ (Watanabe et al., 2018; Karita et al., 2019b), combining the CTC loss (Graves et al., 2006) and attention loss $\mathcal{L}_{ATTN}$, is used to train the speech transformer. The multi-task loss is computed as an interpolation of the two losses with a hyper-parameter $\lambda$ ($0 \leq \lambda \leq 1$):

$$\mathcal{L}_{ATTN} = \text{KL}(\mathbf{p}_{attn} || \mathbf{p}_y), \tag{1}$$

$$\mathcal{L}_{MTL} = \lambda \log \mathbf{p}_{ctc}(\mathbf{y}|\mathbf{h}_{enc}) + (1 - \lambda)\mathcal{L}_{attn}, \tag{2}$$

where $\mathbf{p}_y$ is the label distribution after label smoothing (Müller et al., 2019) to prevent the model from making over-confident predictions. Kullback-Leibler divergence loss (KL) (Kullback & Leibler, 1951), is used for the attention loss.

**Decoding**  Beam search is used to predict the sentence pieces without any additional language models. The decoding score is computed as a weighted sum of both the CTC and attention network probabilities using $\beta$ as the decoding parameter to balance them (Karita et al., 2019a):

$$\hat{\mathbf{y}} = \arg\max_{\mathbf{y} \in \mathcal{Y}^*} \{\beta p_{ctc}(\mathbf{y}|\mathbf{h}_{enc}) + (1 - \beta)p_{attn}(\mathbf{y}|\mathbf{h}_{enc}, \mathbf{y}')\}, \tag{3}$$

where $\mathbf{y}'$ is the decoded sequence so far.

## 2.3 MULTILINGUAL BERT AS TRANSFORMER DECODER

For better language modeling, especially for low-resource languages, mBERT is used as the transformer decoder. Since mBERT is pre-trained on text data, it is essential to augment a cross-attention

layer to the encoder output for each mBERT layer. The cross-attention and its self-attention layers are learned to "align" the acoustic and text spaces for the speech recognition. This is because the text space may diverge significantly from the acoustic space of the encoder output.

**Autoregressive mBERT**  Figure 3 depicts the adaptation of mBERT as an autoregressive transformer decoder. We copy the embeddings and self-attention parameters of mBERT into the decoder layers. Let $t$ denote the current decoding step. The autoregressive decoder takes the current input token $y_t$ to predict the next token $y_{t+1}$. The mBERT embedding layer converts the input token to a vector representation. Subsequently, the cross-attention layer takes the encoder output $\mathbf{h}_{enc}$ as the key and value, and the self-attention output as the query, and computes the attention output.

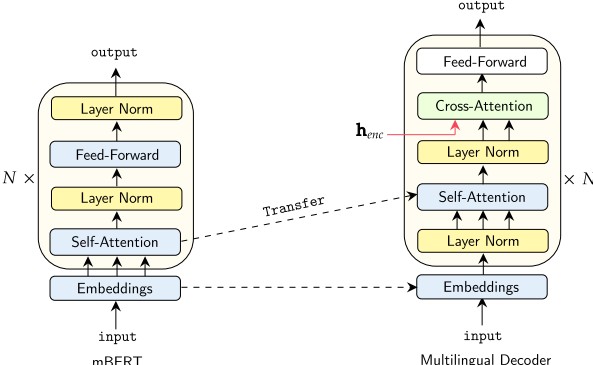

Figure 3: Parameter transfer from a pre-trained multilingual language model to a speech recognition decoder. The dotted line shows the transfer direction from a specific module of the model.

**Vocabulary Mapping**  The vocabulary size of the vanilla mBERT is too large (119,547 tokens) for training the end-to-end speech recognition system. Therefore, vocabulary mapping is performed to reduce the number of targets for the speech transformer. In this work, sentence pieces (SP) (Kudo, 2018) are used as the target tokens. The SP models are trained on the transcriptions with a preset vocabulary size. In this work, we use a shared set of 5,237 tokens as the multilingual system's vocabulary. The minimum number in the token set for the sentence piece model is 150 for all the monolingual systems, except Chinese with 2,265 tokens. The generated sentence piece tokens are then matched against the mBERT token set. During training, the embeddings of all tokens in the mBERT vocabulary are initialized with mBERT embeddings.

## 2.4 Language Adapters

Similar to Kannan et al. (2019), lightweight residual language adapters are used for better acoustic modelling with minimal language-specific parameters to increase the model robustness to languages with limited resources. As shown in Figure 4, in addition to the language-specific adapter for capturing the language-intrinsic knowledge, a common adapter is also trained to learn language-agnostic information in the multilingual data; we call these **Dual-Adapters**. The language-specific and common adapters are denoted as $A_{lang}$ and $A_{com}$, respectively. Each adapter of layer $l$ consists of a down-projection layer $\mathbf{W}_d^l$, followed by a ReLU activation function, and an up-projection layer $\mathbf{W}_u^l$. The adapters take $\mathbf{h}^l$ as the input, where $\mathbf{h}^l$ is the self attention output of layer $l$. We compute the output of $Adapter(\mathbf{h}^l)$ as follows for both the language-specific and common adapters:

$$Adapter(\mathbf{h}^l) = \mathbf{W}_u^l(ReLU(\mathbf{W}_d^l(LayerNorm(\mathbf{h}^l)))) + \mathbf{h}^l. \tag{4}$$

The final adapter output is computed as $\mathbf{o}^l = \mathbf{o}_{lang}^l + \mathbf{o}_{com}^l$. $\mathbf{o}^l$ is then used as the input to the next encoder or decoder layer. We create a language mask to specify the language-specific adapters.

## 2.5 Sentence Piece Class Imbalance Adjustments

The sentence piece class imbalance problem is addressed by incorporating the class priors during training or inference via logit adjustments. Derived from a Bayesian point of view in Menon et al.

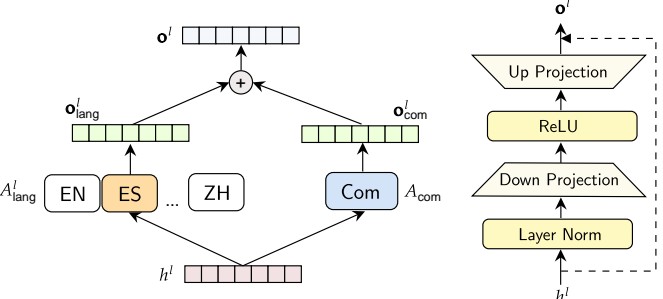

Figure 4: Dual-Adapters. The orange box presents the active language-specific adapter and the blue box presents the common adapter. The figure on the right shows the language adapter structure.

(2020) for computer vision tasks, the softmax classifier with adjusted logits as input minimizes the balanced error across all classes. A natural adjustment is to scale the logits $f_y(x)$ by the inverse of the corresponding class prior $\pi_y$. In log domain, the adjustment can be performed as follows:

$$f_y^{\text{adj}}(x) = f_y(x) - \tau \cdot \log \pi_y, \tag{5}$$

where $\tau > 0$ is a hyper-parameter. The adjustment can be viewed as applying a class-dependent offset to re-weight each logit according to its class prior.

**Class priors** The class priors are the natural frequencies of the sentence piece tokens estimated from the multilingual training data. To form a valid prior distribution, smoothing is applied to the raw counts according to Equation 6 for zero occurrence tokens:

$$\pi_y = \begin{cases} \frac{C_i}{C} - \frac{1}{(N-n_0) \times C}, & c_i > 0 \\ \frac{1}{n_0 \times C}, & \text{otherwise}, \end{cases} \tag{6}$$

where $C$ is the total number of counts for all labels, $n_0$ is the number of labels with zero occurrences, $N$ is the number of classes and $c_i$ is the raw count of class $i$.

**Training phase class imbalance adjustments** To incorporate the priors during training, the logits $f_{y_t}^{\text{dec}}$ of the last decoder layer are adjusted before softmax according to the following:

$$f_{y_t}^{\text{dec}} = w_y^T \cdot \text{Decoder}(\mathbf{h}_{\text{enc}}, \text{Embedding}(y_{t-1})) \tag{7}$$

$$f_{y_t}^{\text{adj}} = f_{y_t}^{\text{dec}} - \tau \cdot \log \pi_{y_t}, \tag{8}$$

$$p_{y_t}^{\text{adj}} = \frac{\exp(f_{y_t}^{\text{adj}})}{\sum_{y_t' \in [N]} \exp(f_{y_t'}^{\text{adj}})}. \tag{9}$$

The adjusted softmax output vector $\mathbf{p}_y^{\text{adj}}$ of the sequence is used to compute the KL loss and perform the backward propagation to update the model. $y_{t-1}$ is the previous label available only during training. To reduce the training and inference discrepancy, scheduled sampling (Bengio et al., 2015) is commonly used for sequential classification tasks like speech recognition. During later training iterations, instead of using the ground truth label $y_{t-1}$ for computing the logits, $y_{t-1}'$ is chosen from the maximum prediction output of the current model to simulate the inference:

$$y_{t-1}' = \underset{y}{\arg\max} \, \mathbf{p}_{y_{t-1}'}^{\text{adj}}. \tag{10}$$

If the scheduled sampling is used, the adjusted logits at step $t$ will have influence over all of the following tokens in the current sequence. This is a crucial difference from the image classification task in Menon et al. (2020). If $\tau$ is set to be 1, the training phase logit adjustment becomes similar to the widely used label smoothing technique Müller et al. (2019). However, in conventional label smoothing, the prior $\pi_y$ is usually a uniform distribution that is independent of the data. The logit adjustment applies a class-specific "smoothing" based on the class prior, and has been shown to be superior to the baseline with the standard label smoothing.

**Inference phase class imbalance adjustments**   Alternatively, the class priors can be incorporated during inference via logit adjustments. The decoding score is computed as follows:

$$\hat{\mathbf{y}} = \arg\max_{\mathbf{y} \in \mathcal{Y}^*} \{\beta \mathbf{p}_{\text{ctc}}(\mathbf{y}|\mathbf{h}_{\text{enc}}) + (1 - \beta)\mathbf{p}_{\mathbf{y}}^{\text{adj}}\}. \tag{11}$$

During beam search, the attention decoding scores $p_{\mathbf{y}}^{\text{adj}}$ are computed in the same way as the scheduled sampling from the adjusted logits.

## 3   EXPERIMENTS

### 3.1   EXPERIMENTAL SETUP

**Dataset**   The CommonVoice dataset (Ardila et al., 2020) is a multilingual corpus collected by Mozilla. Similar to Conneau et al. (2020a), we use 11 languages: *English (en), Spanish (es), French (fr), Italian (it), Kyrgyz (ky), Dutch (nl), Russian (ru), Swedish (sv), Turkish (tr), Tatar (tt),* and *Chinese (zh)*. The dataset is split into training, dev, and eval sets according to the ESPNET recipe. The transcriptions are tokenized using the SentencePiece model with the unigram algorithm. We train our SentencePiece model using speech transcriptions, and the size of the vocabulary is shown in Table 10. We then add special tokens, such as <unk>, <sos>, <eos>, and a blank token for the CTC objective. The detailed data split is shown in Table 6 of Appendix C.

**Network configurations**   We use six transformer encoder layers with a hidden size of 2048 units and eight attention heads, each with an attention dimension of 256. For the decoder, distil-mBERT (Sanh et al., 2019) is used. The mBERT decoder consists of six transformer decoder layers with a hidden size of 3072 and an attention dimension of 756, and four attention heads. We train our model with a batch size of 32 and accumulate the gradient in two steps to have a larger batch size using a single GPU NVIDIA V100 16GB. The models are trained with the Adam optimizer with a warm-up step of 25000. In particular, for balanced sampling, we take six samples for each language and construct a balanced batch by accumulating the gradient 11 times.

**Training and Evaluation**   We evaluate our model using beam-search with a beamwidth of 10 and $\lambda = 0.3$ and $\beta = 0.5$. The hyper-parameter $\tau$ is set to 0.3 for both the training and inference phase class imbalance adjustments. The multilingual models are trained with 150K iterations. We compute the average over the last ten checkpoints as the decoding model. For the monolingual setting, we stop after 100 epochs of training. Models are evaluated using the character error rate (CER) to simplify the evaluation and to have a universal metric for all languages.

**Baselines**   As baseline approaches, we consider the following: **Monolingual**: we train monolingual models; **SMT** (Standard Multilingual Training), we randomly sample the batch from the data distribution; **BS** (Balanced Sampling), we sample the same number of utterances for each language in a batch so that they have roughly equal contributions to the training; **LAN-Specific Adapters**: language-specific adapters proposed by Kannan et al. (2019); and **LID**: (language ID) conditioning with one-hot language vectors proposed by Li et al. (2018).

### 3.2   RESULTS

In Table 1, we present the test results on the CommonVoice dataset. Compared to the monolingual models, even the SMT models improve the performance of the low-resource languages significantly. In other words, SMT is a decent multilingual baseline to be compared with. We conjecture that this may be because the multilingual models can capture common sub-phonetic articulatory features (Kirchhoff et al., 2002; Metze, 2005; Livescu et al., 2007; Wang & Sim, 2014) that are shared by different languages and are beneficial for low-resource languages recognition.

**Balanced Sampling**   We observe the same trend as in Kannan et al. (2019): compared to the SMT, the tail language performance is significantly boosted. However, the performance of the head languages suffers due to fewer occurrences during training. The model is clearly over-fitted to the tail languages due to up-sampling, for example, the CERs on the training set of "ky" and "sv" are significantly lower than the evaluation data (3.4% and 4.2% training vs 13.4% and 22.8% evaluation).

Consequently, the overall performance is the same as SMT. In fact, even after balanced sampling, the sentence piece tokens still have a long-tailed distribution, as shown in Appendix C.

Table 1: Test results in terms of CER (%) on the CommonVoice dataset.

| Model | high-resource | | | intermediate | | | | | low-resource | | | |
|---|---|---|---|---|---|---|---|---|---|---|---|---|
| | en | fr | es | it | ru | zh | tt | nl | tr | ky | sv | avg |
| Training hours | 877 | 273 | 132 | 66 | 57 | 33 | 20 | 19 | 10 | 9 | 4 | |
| Monolingual (full) | 13.3 | 11.5 | 11.1 | 20.7 | 10.3 | 28.2 | 13.6 | 23.0 | 28.0 | 30.0 | 56.1 | 22.3 |
| Training hours | 80 | 50 | 40 | 20 | 15 | 15 | 15 | 15 | 10 | 9 | 4 | |
| Monolingual | 22.6 | 20.1 | 17.3 | 20.7 | 23.9 | 37.6 | 20.7 | 38.1 | 30.3 | 31.6 | 57.7 | 29.1 |
| SMT | **20.1** | **17.4** | **13.0** | **12.5** | **13.7** | **34.1** | **12.2** | **18.7** | **13.9** | **14.4** | **26.4** | **17.9** |
| BS | 25.2 | 20.3 | 14.5 | 12.7 | 13.2 | 32.6 | 11.5 | 18.0 | 12.9 | 13.4 | 22.8 | 17.9 |
| LAN-Specific Adapters | **24.2** | 19.4 | 13.9 | 12.3 | 11.8 | 32.0 | 10.7 | 16.9 | 11.8 | 12.7 | **21.7** | 17.0 |
| LID | 25.7 | 19.2 | 13.7 | 12.0 | 12.0 | 31.6 | 10.8 | 16.4 | 12.0 | 12.5 | 21.8 | 17.1 |
| LID + Adjust-Train | 24.7 | **18.5** | **13.0** | **11.3** | **11.6** | **31.2** | 10.4 | 16.4 | **11.6** | **12.4** | 22.0 | **16.6** |
| A2 (Adjust-Inference) | 22.6 | 17.9 | 12.7 | **11.2** | 11.2 | **30.1** | 10.2 | **15.8** | 11.4 | 12.2 | **21.1** | **16.0** |
| A2 (Adjust-Train) | **22.0** | **17.7** | **12.5** | 11.3 | **11.1** | 30.4 | **10.0** | 15.9 | **11.3** | **12.1** | 21.3 | **16.0** |

**Language Adapters** We next compare the language adaptation techniques, the LAN-Specific Adapters (Kannan et al., 2019), the one-hot language vector (Li et al., 2018), and the Dual-Adapters. Note that all adapters are based on BS + mBERT, which has better performance than the BS-only model. Adding the language-specific adapters without common adapters significantly outperforms the BS baseline, with a 0.9% absolute performance gain. Another way of injecting language information is to augment a one-hot language vector. Interestingly, applying sentence piece class imbalance adjustment (LID + Adjust-Train) to the language vector significantly improves the CER.

**Sentence Piece Class Imbalance Adjustment** Both the training and inference phase adjustments provide a significant performance gain over the LAN-Specific Adapters, with 1% absolute CER reduction. The gains are mostly due to the improved performance of the head languages, although tail languages also benefit from the logit adjustments. More importantly, the gap between the monolingual and multilingual performance for the head languages is greatly reduced, leading to a better "balanced error" performance. This strongly justifies the importance of class imbalance adjustments. Compared to BS, A2 also avoids over-fitting to the tail languages, CERs on "ky" and "sv" are 8.2% and 23.6%, much closer to evaluation CERs. Compared to SMT with random sampling, A2 has a significant better averaged CER with a modest cost for the two head languages "fr" and "en". Some example transcriptions and detailed analysis using different A2 modules are given in Appendix E.

## 3.3 ABLATION STUDIES

### 3.3.1 MULTILINGUAL BERT

The effectiveness of mBERT is presented in Table 2. The performance of mBERT depends heavily on the quality of the acoustic models. Without adapters or logit adjustments, the improvement over BS is marginal, and mBERT performance is even worse for SMT. This may indicate that, with better acoustic models like A2, the text space of the vanilla mBERT is better aligned with the acoustic space, which leads to improved performance across all languages, especially for low-resource ones. It is also interesting to note that, even without adapters, "SMT + mBERT + Adjust-Train" yields the same overall CER as the best adapter system (BS + mBERT + Dual-Adapters).

To study the impacts of the pretrained language models, a more advanced XLM-R (Conneau et al., 2020b) pretrained model is used in place of the distilled-mBERT. Although XLM-R has a better multilingual language generation capability than mBERT, it does not translate to the final performance gain for the multilingual ASR task. We believe this is because it becomes more difficult for the model to align the text and acoustic space with the increased model complexities for XLM-R. XLM-R is not advised considering it has more parameters compared to distilled-mBERT, and the performance gain is not significant although it does improve the performance on "zh" and "fr" slightly.

Table 2: Ablation study on mBERT for multilingual speech recognition.

| | en | fr | es | it | ru | zh | tt | nl | tr | ky | sv | avg |
|---|---|---|---|---|---|---|---|---|---|---|---|---|
| SMT | **20.1** | **17.4** | **13.0** | **12.5** | **13.7** | 34.1 | **12.2** | 18.7 | **13.9** | 14.4 | 26.4 | **17.9** |
| SMT + mBERT | 20.4 | 17.9 | 13.4 | 13.0 | 14.0 | 34.8 | 12.4 | 18.9 | 14.0 | **14.3** | **26.2** | 18.1 |
| BS | 25.2 | 20.3 | **14.5** | 12.7 | **13.2** | 32.6 | 11.5 | 18.0 | 12.9 | 13.4 | **22.8** | 17.9 |
| BS + mBERT | **25.0** | **20.0** | 15.4 | 12.6 | **13.2** | 32.9 | **11.1** | 17.3 | 12.6 | 12.8 | **22.8** | 17.8 |
| BS + Dual-Adapters | 23.5 | 18.9 | 13.5 | 12.1 | 12.3 | 31.0 | 10.9 | 16.5 | 12.0 | 12.9 | 21.6 | 16.8 |
| BS + Dual-Adapters + mBERT | **23.4** | **18.6** | **13.4** | **11.8** | **11.7** | **30.8** | **10.8** | **16.2** | **11.6** | 12.4 | 21.5 | **16.5** |
| SMT + Adjust-Train | 20.2 | 16.9 | 12.5 | 11.9 | 13.1 | 32.9 | 11.3 | 18.5 | 13.5 | 13.9 | 25.3 | 17.3 |
| SMT + Adjust-Train + mBERT | **19.4** | **16.5** | **12.1** | **11.7** | **12.2** | **31.1** | **11.0** | 17.5 | 12.7 | 13.2 | 24.6 | **16.5** |
| A2 with mBERT | **22.0** | 17.7 | **12.5** | 11.3 | 11.1 | 30.4 | **10.0** | 15.8 | 11.3 | 12.1 | 21.3 | **16.0** |
| A2 with XLM-R | 22.1 | **17.6** | **12.5** | 11.4 | 11.4 | **29.6** | 10.3 | 15.9 | 11.5 | **12.1** | 21.4 | **16.0** |

### 3.3.2 LANGUAGE ADAPTERS

The results and parameter sizes of different adapters are given in Table 3. Generally speaking, decoder layer adapters are not as effective as in the encoder layers, indicating adaptation of the acoustic space is much more effective than of the text space. Therefore, considering the extra computation and parameters, it is advisable to apply only the encoder adapters.

Table 3: Ablation study on language adapters' impacts on BS + mBERT models

| | #Params | en | fr | es | it | ru | zh | tt | nl | tr | ky | sv | avg |
|---|---|---|---|---|---|---|---|---|---|---|---|---|---|
| No Adapter | 76M | 25.0 | 20.0 | 15.4 | 12.6 | 13.2 | 32.9 | 11.1 | 17.3 | 12.6 | 12.8 | 22.8 | 17.8 |
| Decoder Adapter Only | 82M | 23.6 | 19.2 | 13.7 | 12.2 | **11.7** | 32.5 | **10.3** | 16.7 | 11.7 | **12.2** | 21.4 | 16.8 |
| Encoder Adapter Only | 78M | **23.1** | **18.5** | 13.3 | 11.8 | **11.7** | 31.1 | 10.6 | **16.0** | **11.5** | 12.6 | **21.3** | **16.5** |
| Encoder + Decoder | 84M | 23.4 | 18.6 | 13.4 | **11.8** | **11.7** | **30.8** | 10.8 | 16.2 | 11.6 | 12.4 | 21.5 | **16.5** |

We investigate the effectiveness of the common language adapters in Table 4. The Dual-Adapters outperform the language-specific adapters significantly, by a 0.5% absolute CER reduction, indicating knowledge transfer with the common adapter is effective.

Table 4: Ablation study of language adapters.

| | #Params | en | fr | es | it | ru | zh | tt | nl | tr | ky | sv | avg |
|---|---|---|---|---|---|---|---|---|---|---|---|---|---|
| LAN-Specific Adapters | 84M | 24.2 | 19.4 | 13.9 | 12.3 | 11.8 | 32.0 | 10.7 | 16.9 | 11.8 | 12.7 | 21.7 | 17.0 |
| Individual Dual-Adapters | 84M | **23.4** | **18.6** | **13.4** | **11.8** | 11.7 | **30.8** | 10.8 | **16.2** | **11.6** | 12.4 | 21.5 | **16.5** |
| Language Group Dual-Adapters | | | | | | | | | | | | | |
| By Written Scripts | 78M | 24.0 | 19.4 | 13.9 | 12.1 | 11.7 | 32.1 | 10.5 | 16.3 | 12.0 | 12.0 | **21.0** | 16.8 |
| By Language Families | 78M | 23.5 | 19.0 | 13.6 | 11.9 | **11.4** | 31.0 | **10.4** | 16.2 | 11.6 | 11.9 | 21.4 | **16.5** |

In addition to the individual language adapters, we also divide languages into groups to allow sharing of adapters within the same group. According to the written scripts, we divide the 11 languages into language groups, *e.g.,* Latin, Chinese characters and Cyrillic scripts. They can also be groups into language families, *e.g.,* Romance, Chinese, Turkic, Germanic. This group focuses more on the similarities in lexica, grammars, and pronunciations, which are usually subsumed under the end-to-end multilingual architectures. According to one group, languages that belong to the same cluster do not necessarily belong to the same cluster in the other group. For example, Tartar and Turkish are both Turkic languages. However, Tartar uses the Cyrillic script, and Turkish uses the Latin alphabet. All languages in the same group share the same dual-adapters, and the adapters are trained with all language members. In general, grouping by language families is better than grouping by written scripts because it is more consistent with the encoder adapters for adapting the acoustic space, which are more effective than decoder adapters in Table 3. Compared to individual language adapters, sharing language adapters by language families helps the low-resource languages performance, *e.g.,* "sv" of the Germanic group, "ky" and "tr" of the Turkic group because more data are used to train the group adapters. However, this also comes with a cost to the resource-rich languages compared to "Individual Dual-Adapters". Therefore, individual language adapters are advised considering the adapters' parameter sizes are much smaller than the encoder and decoder attention weights.

### 3.3.3 SENTENCE PIECE CLASS-IMBALANCE ADJUSTMENTS: TRAINING VS. INFERENCE

The two logit adjustments are compared in Table 5. For the SMT systems, training phase adjustment shows a clear advantage over inference phase adjustment. Under the convex assumption, the solution of the two adjustment approaches is the same. However, deep neural network optimization is a non-convex problem, so they may converge to different local minima. Under SMT, the model is heavily biased towards the head classes due to random sampling. Training phase class imbalance adjustment can help the training to place more focus on the tail classes, leading to much better balanced and lower error. With better acoustic models, *e.g.,* language adapters, the inference phase adjustment can better calibrate the raw classification scores and yield similar performance to the training phase adjustment. Lastly, we show the effects of $\tau$ on inference phase logit adjustment in Appendix D.

Table 5: Training and inference phase logit adjustments with different A2 models.

| | Params | en | fr | es | it | ru | zh | tt | nl | tr | ky | sv | avg |
|---|---|---|---|---|---|---|---|---|---|---|---|---|---|
| SMT + mBERT + | 76M | 20.4 | 17.9 | 13.9 | 13.0 | 14.0 | 34.1 | 12.4 | 18.9 | 14.0 | 14.3 | 26.2 | 18.1 |
| Adjust-Inference | 76M | 20.1 | 17.2 | 13.4 | 12.2 | 13.2 | 34.0 | 11.5 | 18.3 | 13.4 | **13.2** | 26.1 | 17.6 |
| Adjust-Train | 76M | **19.4** | **16.5** | **12.1** | **11.7** | **12.2** | **31.1** | **11.0** | **17.5** | **12.7** | 13.2 | **24.6** | **16.5** |
| BS + mBERT + | 76M | 25.0 | 20.0 | 15.4 | 12.6 | 13.2 | 32.9 | 11.1 | 17.3 | 12.6 | 12.8 | 22.8 | 17.8 |
| Adjust-Inference | 76M | 24.1 | 19.3 | 13.5 | 12.0 | 12.1 | 32.3 | 10.6 | **16.7** | 12.2 | **12.6** | 22.2 | 17.0 |
| Adjust-Train | 76M | **23.8** | **19.1** | **13.4** | **11.9** | **11.8** | 32.4 | 10.5 | 16.7 | **12.1** | 12.6 | **21.5** | **16.9** |
| BS + mBERT + Dual-Adapters + | 84M | 23.4 | 18.6 | 13.4 | 11.8 | 11.7 | 30.8 | 10.8 | 16.2 | 11.6 | 12.4 | 21.5 | 16.5 |
| Adjust-Inference | 84M | 22.6 | 17.9 | 12.7 | **11.2** | 11.2 | **30.1** | 10.2 | **15.8** | 11.4 | 12.2 | **21.1** | **16.0** |
| Adjust-Train | 84M | **22.0** | **17.7** | **12.5** | 11.3 | **11.1** | 30.4 | **10.0** | 15.9 | **11.3** | 12.1 | 21.3 | **16.0** |

## 4 RELATED WORK

**Long-Tail** Conventional approaches to addressing the long-tail problem are focused on data sampling methods (Kubat & Matwin, 1997; Chawla et al., 2002; Wallace et al., 2011). Recently, the long-tail distribution issue has regained interest for neural network models (Menon et al., 2020), and several approaches have been proposed, such as weight normalization (Kang et al., 2019), adaptive margin (Cao et al., 2019), and equalized loss (Tan et al., 2020).

**Adapters** Adapters were first proposed to learn domain-specific representations in computer vision in a parameter-efficient way (Rebuffi et al., 2017). They were subsequently adopted for NLP tasks to avoid fine-tuning a new model for each task by training an adapter module for each task while sharing the pre-trained language model parameters (Houlsby et al., 2019; Lin et al., 2020). Invertible adapters were proposed in Pfeiffer et al. (2020a;b) to effectively adapt an existing pre-trained multilingual model to unseen languages for multi-task cross-lingual transfer.

**Multilingual ASR** E2E architectures like LAS (Toshniwal et al., 2018) and the Recurrent Neural Transducer (Kannan et al., 2019) have been used for building a multilingual ASR system for a group of Indian languages. In Kannan et al. (2019), language adapters are used to tackle the data imbalance problem, although the improvement from using language adapters is marginal compared to the language vector augmentation. Acoustic vector quantization is also used in the recent work by Conneau et al. (2020a) on multilingual ASR. A massive multilingual ASR study with more than 50 languages and more than 16,000 hours of speech is presented in Pratap et al. (2020). The two main techniques are data resampling and language clusters, which bear some similarities with our balanced sampling and language adapters. Unfortunately, their datasets are not publicly available.

## 5 CONCLUSION

In this paper, we introduce Adapt-and-Adjust (A2), an end-to-end transformer-based framework to address the crucial challenge of long-tail data distribution issues in multilingual speech recognition. A2 consists of three major components, namely, language adapters, class imbalance adjustments, and a pretrained multilingual language model. Extensive experiments on the CommonVoice corpus show that A2 significantly outperforms conventional approaches for end-to-end multilingual speech recognition due to 1) the better acoustic modeling with adapters and class imbalance adjustments; and 2) the better language modeling with pretrained multilingual BERT.

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

## A    A2 ALGORITHM

The full algorithm of the A2 framework with training phase sentence piece class imbalance adjustments is outlined in Algorithm 1.

---

**Algorithm 1** Adapt-and-Adjust (A2) with Adjust-Train

---

**Require:** $\mathcal{D}$: long-tailed multilingual data
**Require:** $\theta$: an encoder-decoder model
**Require:** $\alpha$, $\lambda$: step size hyperparameters

1: randomly initialize $\theta$
2: copy mBERT pretrained language model to decoder parameters in $\theta$
3: compute class priors $\pi$ from $\mathcal{D}$
4: **while** not done **do**
5:     Sample batch of multilingual utterances $\mathbf{x} \sim \mathcal{D}$
6:     Generate language adapter mask $\mathbf{m}$ using the language tag in $\mathbf{x}$
7:     Compute encoder hidden states $\mathbf{h}_{\text{enc}}$ using $\mathbf{x}$ and $\mathbf{m}$ by encoder forwarding
8:     Compute logits $f_{y_t}^{\text{dec}}$ using $\mathbf{h}_{\text{enc}}$ and $\mathbf{m}$ by decoder forwarding
9:     Adjust logit according to Equation 9
10:     Compute CTC posteriors $p_{\text{ctc}}(\mathbf{y}|\mathbf{h}_{\text{enc}})$
11:     Compute attention loss $\mathcal{L}_{\text{ATTN}}$ in Equation
12:     Compute multi-task loss $\mathcal{L}_{\text{MTL}}$ using $p_{\text{ctc}}(\mathbf{y}|\mathbf{h}_{\text{enc}})$, $\mathcal{L}_{\text{ATTN}}$, and $\lambda$ in Equation 2
13:     Update model $\theta \leftarrow \theta - \alpha \nabla_\theta \mathcal{L}_{\text{MTL}}$
14: **end while**

---

## B    ASR MODEL STRUCTURE

The encoders and decoders of the transformer-based speech recognition model used in this study is presented here.

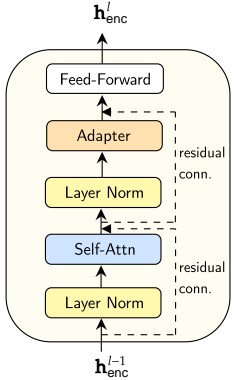
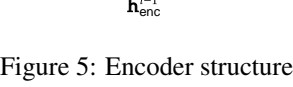

Figure 5: Encoder structure.

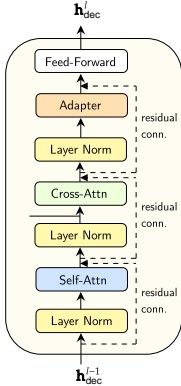

Figure 6: Decoder structure.

**Encoder Layer**    Figure 5 shows the structure of the encoder layer. An adapter layer is added after the layer norm and self-attention module. We also apply two residual connections after both the self-attention layer and the adapter layer:

$$\mathbf{o} = \text{SelfAttn}(\text{LayerNorm}(\mathbf{h}_{\text{enc}}^{l-1})) + \mathbf{h}_{\text{enc}}^{l-1}, \tag{12}$$

$$\mathbf{h}_{\text{enc}}^{l} = \text{FeedForward}(\text{Adapter}(\text{LayerNorm}(\mathbf{o})) + \mathbf{o}), \tag{13}$$

where $\mathbf{h}_{\text{enc}}^{l-1}$ is the encoder hidden states of the previous layer $l-1$ and $\mathbf{h}_{\text{enc}}^{l}$ is the output of the encoder layer.

**Decoder Layer** Figure 6 shows the structure of decoder layer with the adapter. We place the adapter layer after the cross-attention model.

$$\mathbf{o}_1 = \text{SelfAttn}(\text{LayerNorm}(\mathbf{h}_{\text{dec}}^{l-1})) + \mathbf{h}_{\text{dec}}^{l-1}, \tag{14}$$

$$\mathbf{o}_2 = \text{CrossAttn}(\mathbf{h}_{\text{enc}}, \text{LayerNorm}(\mathbf{o}_1)) + \mathbf{o}_1 \tag{15}$$

$$\mathbf{h}_{\text{dec}}^{l+1} = \text{FeedForward}(\text{Adapter}(\text{LayerNorm}(\mathbf{o}_2)) + \mathbf{o}_2), \tag{16}$$

where $\mathbf{h}_{\text{dec}}^{l-1}$ is the decoder hidden states of the previous layer, and $\mathbf{h}_{\text{dec}}^{l}$ is the output the current layer.

## C   DATASET DISTRIBUTION

Table 6 shows the data split used for training, validation, and testing. Table 7 shows the frequencies of the labels in the original data and our estimates on the balance sampling. There are only a few data with an occurrence of more than 50k (around 1%). The class distribution on the balance sampling is shifted to the 10k-50k range; thus, there are more labels in the middle of the distribution.

Table 6: Dataset split information

|  | high-resource | | | intermediate | | | | | low-resource | | |
|---|---|---|---|---|---|---|---|---|---|---|---|
|  | en | fr | es | it | ru | zh | tt | nl | tr | ky | sv |
| Train (hours) | 80.0 | 50.0 | 40.0 | 20.0 | 15.0 | 15.0 | 15.0 | 10.0 | 10.0 | 9.0 | 4.0 |
| Valid (hours) | 109.0 | 34.0 | 16.5 | 8.3 | 7.3 | 4.2 | 2.5 | 2.5 | 1.3 | 1.1 | 0.5 |
| Test (hours) | 10.0 | 10.0 | 10.0 | 8.3 | 7.3 | 4.3 | 2.5 | 2.5 | 1.3 | 1.1 | 0.5 |

Table 7: Class frequency in the original data and after applying balanced sampling

|  | >100k | 50k-100k | 10k-50k | 1k-10k | <1k |
|---|---|---|---|---|---|
| Original data | 6 | 9 | 152 | 1164 | 3905 |
| Balance sampling | 12 | 30 | 306 | 960 | 3243 |

Figure 7 and Figure 8 visualizes the long-tail distribution in more detailed. To better visualize them, we ignore the classes fewer than 20k.

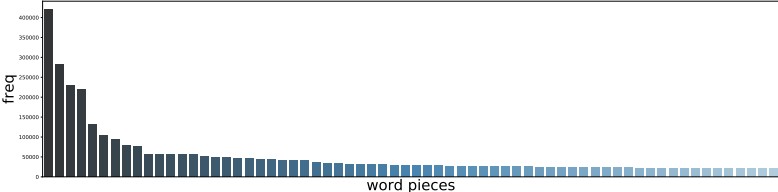

Figure 7: Class Frequency. We only show labels more than 20k

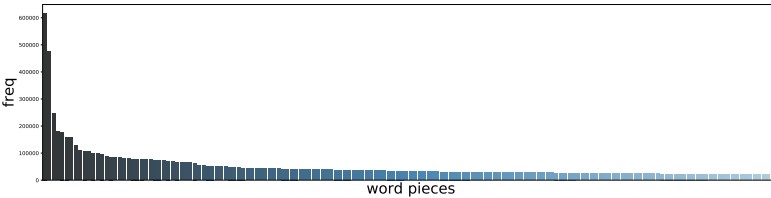

Figure 8: Class Frequency with Balance Sampling. We only show labels more than 20k

## D ABLATION STUDY ON INFERENCE PHASE LOGIT ADJUSTMENT

Figure 9 depicts the CERs of the systems with different $\tau$ for inference phase logit adjustment with the best A2 configuration: balanced sampling with mBERT and Dual-Adapters. We choose one language from each group, and the other languages demonstrate the same trend. In Figure 9, the advantage of the logit adjustment is clearly shown compared to the baseline without logit adjustment ($\tau = 0$). All of the languages achieve their best CER with a $\tau$ value of $\tau = [0.3, 0.4]$. Performance saturates with heavier smoothing ($\tau >= 0.5$).

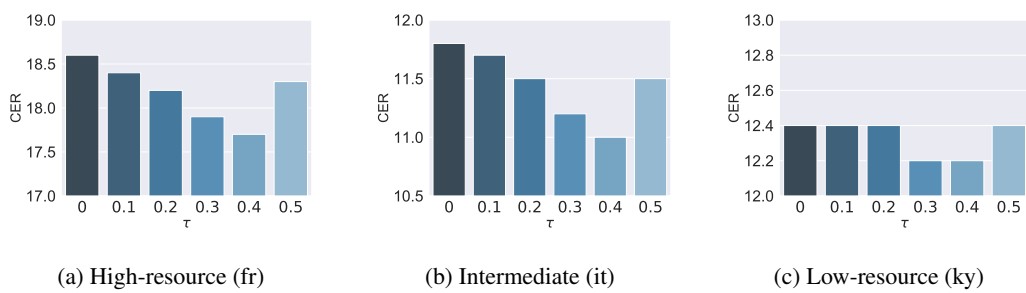

| (a) High-resource (fr) | (b) Intermediate (it) | (c) Low-resource (ky) |

Figure 9: Comparison of models with different $\tau$.

## E TRANSCRIPTION EXAMPLES

We provide samples of two languages generated from different models, a resource-limited language *zh* (traditional Chinese scripts and Mandarin spoken in Taiwan) and a resource-rich language *en* (English).

Table 8: Transcription of a testing traditional Chinese utterance with different models

| Reference | 困難與挑戰是激發我們的原動力 | |
| Pinyin | Kun4 Nan2 Yu3 Tiao3 Zhan4 Shi4 Ji1 Fa1 Wo3 Men2 De0 Yuan2 Dong4 Li4 | |
| **Model** | **Hypothesis** | **CER** |
| BS + mBERT | 困難與挑戰是資料我們的員動力 | 21.4 |
| + Dual-Adapters | 困難與挑戰是機發我們的員動力 | **14.3** |
| + Dual-Adapters + Adjust-Train | 困難與挑戰是機發我們的員動力 | **14.3** |
| Monolingual | 負能一票佔是機發我的能員動力 | 64.3 |
| SMT | 可能與調站是機發我們的員動力 | 42.9 |

The reference in Table 8 roughly translates to English as "Obstacles and challenges are the sources of power that motivate us." The monolingual output is rather poor. Not only did it miss an important character 們 for distinguishing "my" and "our", the sounding out of the sentence is also quite different from the reference (Fu4 Neng2 Yi2 Piao4 Zhan4 Shi4 Wo3 De0 Yuan2 Dong4 Li4). The standard multilingual training (SMT) improves the monolingual training by predicting the character 與 (Yu3) in place of 一 (Yi2). In addition, the output of 調站 (Tiao2 Zhan4) sounds almost the same as the reference 挑戰 (Tiao3 Zhan4). (The monolingual system output 票佔 (Piao4 Zhan4) not only does not make any sense in terms of word meaning, the pronunciation is also quite different from the reference.) This shows that the multilingual training helps improve the acoustic modelling compared to the monolingual training if the monolingual training data are limited. The mBERT model further helps the language modeling by correcting 可能與調站 (literally translates to "possibility and changing stations") to 困難與挑戰 ("obstacles and challenges"). However, mBERT also replaces 機發 (Ji1 Fa1) to 資料 (Zi1 Liao4) since 資料 ("documents/materials") is a more common word than 機發, which is a wrong word with correct pronunciations relative to the reference. Lastly, the adapters successfully convert 資料 (Zi1 Liao4) to 機發 (Ji1Fa1) so that it sounds the same as the reference with one wrong character. This demonstrates the adapter's capability in better acoustic modelling. Finally the errors including the wrong characters in 機發 (Ji1Fa1) and 員動力 (Yuan2

Dong4 Li4) can be easily corrected by a decent external language model during decoding, *e.g.,* an RNN-LM trained on the training transcriptions.

Table 9: Transcription of a testing English utterance with different models

| Reference | FOUND A LITTLE CROWD OF ABOUT TWENTY PEOPLE SURROUNDING THE HUGE HOLE | |
| --- | --- | --- |
| **Model** | **Hypothesis** | **CER** |
| BS + mBERT | I SOUND THE LITTLE CROWD OF THE ABOUT TWENTY PEOPLE SURROUNDING THE HUGE HOLE | 13.6 |
| + Dual-Adapters | I SOUND A LITTLE CROWD IS ABOUT SPENT PEOPLE SURROUNDING THE HUGE HOLE | 11.9 |
| + Dual-Adapters + Adjust-Train | I SOUND THE LITTLE CROWD OF ABOUT TWENTY PEOPLE SURROUNDING THE HUGE HOLE | **8.5** |
| Monolingual | SAW THE LEVEL CROWD AS ABOUT TWENTY PEOPLE SURROUNDING THE FUISH FOR | 33.9 |
| SMT | SOUND THE LITTLE CROWD AS ABOUT TWENTY PEOPLE SURROUNDING THE HUGE HOLE | 13.6 |

Table 9 shows the transcriptions of different models applied to an English testing utterance. Different from Table 8, where only limited training data are available, mBERT does not provide any improvement over SMT, except that it makes the sentence more grammatical by adding a subject, "I". The Dual-Adapters improve the performance compared to mBERT by successfully decoding the sentence piece "ENT" in "SPENT", which is closer to "TWENTY" in terms of characters. Lastly, the class imbalance adjustment system yields the best CER.

# F    VOCABULARY COVERAGE

Table 10: Vocabulary coverage

| | high-resource | | | intermediate | | | | | low-resource | | | |
| --- | --- | --- | --- | --- | --- | --- | --- | --- | --- | --- | --- | --- |
| | **en** | **fr** | **es** | **it** | **ru** | **zh** | **tt** | **nl** | **tr** | **ky** | **sv** | **multi** |
| Coverage (%) | 74.4 | 73.6 | 70.8 | 73.9 | 90.2 | 97.7 | 91.3 | 64.7 | 57.1 | 91.1 | 53.8 | 54.9 |
| # Match | 181 | 281 | 192 | 164 | 156 | 2,330 | 157 | 110 | 93 | 154 | 86 | 2,875 |
| # Vocab | 243 | 382 | 271 | 222 | 173 | 2,386 | 172 | 170 | 163 | 169 | 160 | 5,237 |

# G    LOSS DYNAMICS

We plot the validation losses from A2 and the other baselines in Figures 10 and 11.

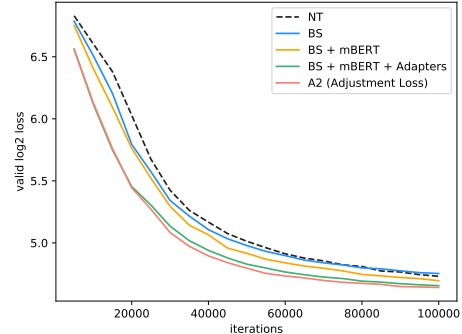

Figure 10: Validation losses.

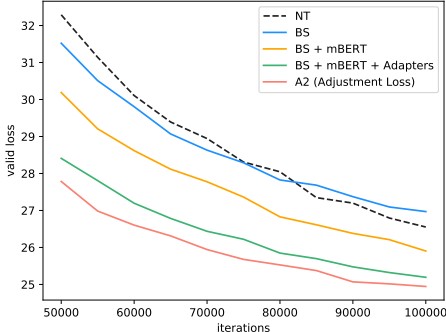

Figure 11: Closer look at validation losses.

# H    LANGUAGE GROUP ADAPTERS

The language groups used in the experiment are shown in Table 11. We also present the detailed CER results in Table 12 with sentence piece class imbalance adjustments. The inference phase logit adjustments is slightly better than the training phase adjustment.

Table 11: Language groups used in the adapters.

| Groups | Languages |
|---|---|
| **By Written Scripts** | |
| Latin | en, fr, es, it, nl, tr, sv |
| Chinese | zh |
| Cyrillic | ru, tt, ky |
| **By Language Families** | |
| Germanic | en, nl |
| Romance | fr, es, it, sv |
| Turkic | tr, tt, ky |
| Chinese | zh |

Table 12: Character error rate (CER) using different language group adapters.

| | high-resource | | | intermediate | | | | | low-resource | | | |
|---|---|---|---|---|---|---|---|---|---|---|---|---|
| | en | fr | es | it | ru | zh | tt | nl | tr | ky | sv | avg |
| Language Families + | 23.5 | 19.0 | 13.6 | 11.9 | 11.4 | 31.0 | 10.4 | 16.2 | 11.6 | 11.9 | 21.4 | 16.5 |
| Adjust-Train | 23.1 | 18.4 | 13.2 | 11.4 | 11.3 | 31.5 | 10.1 | 16.1 | 11.7 | 12.4 | **20.6** | 16.3 |
| Adjust-Inference | **22.7** | **18.2** | **12.9** | **11.3** | **11.1** | **30.7** | **10.0** | **15.9** | **11.5** | **11.8** | 20.7 | **16.1** |
| Language Scripts + | 24.0 | 19.4 | 13.9 | 12.1 | 11.7 | 32.1 | 10.5 | 16.3 | 12.0 | 12.0 | 21.0 | 16.8 |
| Adjust-Train | 23.5 | 18.9 | 13.5 | 11.7 | 11.6 | **31.5** | 10.4 | 16.4 | 11.9 | 12.1 | 21.2 | 16.6 |
| Adjust-Inference | **23.3** | **18.6** | **13.1** | **11.5** | **11.5** | 31.5 | **10.0** | **16.0** | **11.8** | **11.8** | 20.8 | **16.4** |

## I  FEW-SHOT SETTING

We also investigate the effectiveness of the A2 framework in the few-shot setting. We take five-hour speech data of each language as the training data. The test results are shown in Table 13. Using mBERT as the speech decoder consistently improves the performance in the low-resource monolingual scenario. Interestingly, for Chinese language "zh", without using mBERT as the decoder, the model seems to be unable to learn useful information and the mBERT decoder yields an absolute CER reduction of 17.2%.

For multilingual training, marginal performance gains are observed with mBERT decoder, indicating the multilingual language model is complementary to the shared acoustic knowledge learned from the multilingual speech data. The language adapters lead to consistent and significant performance gain across all languages. Note the sentence piece class imbalance adjustments do not yield significant performance boost as in the long-tail multilingual setting in the main text. We believe this is largely due to less severe sentence piece class imbalance when the same amount of training data for each language is used.

Table 13: Character error rate (CER) on five-hour CommonVoice training data

| | high-resource | | | intermediate | | | | | low-resource | | | |
|---|---|---|---|---|---|---|---|---|---|---|---|---|
| | en | fr | es | it | ru | zh | tt | nl | tr | ky | sv | avg |
| *Monolingual* | | | | | | | | | | | | |
| Standard Training (ST) | 69.6 | 61.1 | 56.5 | 49.8 | 49.6 | 96.4 | 32.8 | 57.5 | 49.6 | 43.7 | 56.1 | 56.5 |
| ST + mBERT | 67.3 | 56.1 | 51.0 | 45.4 | 42.8 | 73.2 | 29.6 | 52.9 | 46.1 | 39.6 | 55.1 | 50.8 |
| *Multilingual* | | | | | | | | | | | | |
| SMT | 41.4 | 33.5 | 25.2 | 22.3 | 21.9 | 43.0 | 15.5 | 24.9 | 18.7 | 18.9 | 28.9 | 26.7 |
| SMT + mBERT | 40.9 | 32.6 | 24.3 | 21.0 | 19.9 | 42.1 | 14.5 | 23.8 | 17.3 | 17.4 | 26.7 | 25.5 |
| SMT + mBERT + Adapters | 36.7 | 28.6 | 21.3 | 18.3 | 17.2 | 38.2 | 12.9 | 20.9 | 15.1 | 15.1 | 23.9 | 22.6 |
| A2 (Adjust-Inference) | **36.5** | **28.4** | **21.0** | **18.0** | **16.9** | **38.0** | **12.7** | **20.6** | **14.8** | **15.0** | **23.7** | **22.3** |

## J    HYPER-PARAMETER SETTING

The hyper-parameters used in the training are shown in Table 14.

Table 14: Hyper-parameter setting for training

| hyper-parameters | value |
|---|---|
| encoder-layer | 6 |
| encoder-units | 2048 |
| decoder-layers | 6 |
| decoder-units | 3072 |
| encoder-attention-dim | 256 |
| decoder-attention-dim | 756 |
| attention-heads | 4 |
| label-smoothing | 0.1 |
| batch-size | 32 (for random sampling) |
| | 6 (for balanced sampling) |
| maximum-length-in | 512 |
| maximum-length-out | 150 |
| optimizer | noam |
| warmup-step | 25000 |
| accum-grad | 2 (for random sampling) |
| | 11 (for balanced sampling) |
| grad-clip | 5 |

