# OpenReview forum: "Adapt-and-Adjust: Overcoming the Long-tail Problem of Multilingual Speech Recognition"
_ICLR.cc/2021/Conference — Reject_

### Official Review · AnonReviewer2 · 2020-10-28
**Review comments for paper 3487**

**Rating:** 5
**Confidence:** 4

**Review:**

This paper proposes an Adapt-and-Adjust framework to address the long-tail problem in multilingual ASR, which assembles three techniques: 1) leveraged a pre-trained model mBERT to initialize the decoder, 2)  language-specific and language-agnostic adaptors, 3) class imbalance adjustments.  Experiments on a multilingual ASR with 11 languages demonstrate the proposed method can achieve accuracy improvements.

Overall this paper is clearly written and easy to follow. Each technique is presented with details and evaluated with corresponding ablation studies. It is a good paper in terms of application, experiments and systematic engineering efforts.  However, I have several concerns on the overall novelty and technical contributions:
1) The three techniques alone are not novel enough, and each is proposed by previous works. E.g., initialized with a pre-train language model, class imbalance adjustment, and language-specific adaptors which are similar to mixture of language experts.
2) The proposed method can hardly be called as a framework since it has not demonstrated its necessity and applicability for each component. In another view, it is more like an assemble of different improvement tricks without much centralized logic towards a dedicated and focused problem.
3) The effectiveness of a component (mBERT) need to depend on other components, otherwise it does not work. This makes the proposed method not generalizable. Why mBERT is only effective when coupled with others? Is it necessary? Is the improvement by chance but not universal?
4) Initializing from mBERT (trained with MLM) but adjusting to autoregressive generation would harm the model capability of mBERT. Why not initialize from GPT model or more appropriate from sequence to sequence pre-trained models with an cross-attention module such as MASS or BART? This would be more effectiveness than simply using mBERT.

---

> ### Author Response · Authors · 2020-11-19
> **Responses to Reviewer 2**
>
> Thank you for the constructive review. We have updated our paper to address your comments. We would answer your questions and concerns as follows:
>
> **Q: The three techniques alone are not novel enough, and each is proposed by previous works. E.g., initialized with a pre-train language model, class imbalance adjustment, and language-specific adapters, which are similar to a mixture of language experts.**
>
> - Thanks for the comment. Initialization with a pre-trained language model is not new in NLP, but to our best knowledge, we are the first who propose this to the speech recognition task, and it’s non-trivial to get it to work for ASR. Class imbalance adjustment was mainly addressed in the computer vision tasks, which is a classification task. In this paper, we apply the class imbalance adjustment to a sequential classification task, i.e., ASR, with an autoregressive decoder.
> - Language-specific adapters are not the same as language experts. Adapters are light-weight parameters that are added to encoder and decoder layers to learn language-specific information. They reside in a single model and can be easily replaced with new tasks without changing the other model parameters. On the other hand, to our understanding, a mixture of language experts are basically utilizing multiple encoders or decoders and mix the output of the experts. Language-specific adapters were firstly used by Google in paper [1], here we improve the adapters by adding a common adapter (the Dual-Adapters) to encode the shared knowledge between languages.
>
> [1] Kannan, et al. Large-scale multilingual speech recognition with a streaming end-to-end model. Interspeech.
>
> **Q: The proposed method can hardly be called as a framework since it has not demonstrated its necessity and applicability for each component. In another view, it is more like an assemble of different improvement tricks without much-centralized logic towards a dedicated and focused problem.**
>
> - We want to emphasize that the primary problem we are addressing is to train a robust multilingual ASR with a single end-to-end model to improve the recognition of low-resource languages while keeping the recognition performance for the high-resource languages, compared to the monolingual models or the standard multilingual training.
> The first challenge we addressed is the long-tail class distribution problem, where we have demonstrated the advantages of applying the logit adjustment alone in table 2. Logit adjustment is a must-have component to address the long-tail label distribution problem.
> - Another challenge from multilingual ASR is the lack of training data for certain languages. To solve the data scarcity problem, pre-trained mBERT is used for better language modeling, the Dual-adapters are used for better acoustic modeling by adapting the models learned from the pooled training data to the low-resource languages.
>
> **Q: The effectiveness of a component (mBERT) need to depend on other components; otherwise, it does not work. This makes the proposed method not generalizable. Why mBERT is only effective when coupled with others?**
>
> Since mBERT is trained on the text data only, the text space may not be consistent with the transcriptions of the ASR training data. In order to use it for the ASR decoder, its text space needs to be aligned with the acoustic space in the ASR encoder. We believe the reason why it needs to be coupled with others is that the better acoustic models with dual-adapters or logit adjustment help the alignment between the text space of mBERT with the acoustic space with the encoder via cross-attentions. Nevertheless, the improvement is not as significant as the other two techniques, namely dual adapter and logit adjustments. Therefore, mBERT is probably the only component that can be replaced for the sake of model size and computation requirements.
>
> **Q: Why not initialize from GPT model or more appropriate from sequence to sequence pre-trained models with a cross-attention module such as MASS or BART?**
>
> Thanks for the comment, as presented in the previous comment, the improvement of mBERT is not as significant as the other two techniques. Although using larger models like mBART would be more principled, it increases the model size drastically, which will require a much larger GPU memory size and slow down the training significantly. Most importantly, it will affect the decoding speed, making it less practical. In fact, we did try using mBART, unfortunately, the model can barely fit into our GPU memory, and we can only use a batch size of 2. Considering the improvement is small, we did not proceed. Nevertheless, we have trained a new model with a more advanced pretrained language model XLM-R as the decoder to show the effect of a bigger pre-trained language model in the revised paper. The results are presented in Table 2, adding the more advanced pre-trained language model does not provide ASR performance gain in terms of CER.

---

### Official Review · AnonReviewer1 · 2020-10-29
**A large disconnect between the proposed additions and the problems the paper tries to solve**

**Rating:** 4
**Confidence:** 4

**Review:**

The paper proposes three additions to improve a monolithic multilingual end-to-end ASR system. The problem of training a monolithic multilingual ASR system is that using data from multiple languages does not necessary improve over individual monolingual systems. The three additions are a large multilingual language model, the use of language adapters, and smoothing on the token probabilities. Mixing the three additions in a specific way helps improve the average word error rates.

There are two major problems in the paper. One is the imprecise use of words, and the other is the disconnect between the additions and the problems they try to solve. Details are as follows.

The paper contains a lot of imprecise use of words. The term "long tail" is used throughout the paper, but it is never clearly defined. The long tail of a distribution refers to a significant total amount of probability mass spread on a large support. In the context of this paper, when the paper talks about the long-tail problem, what distribution are we talking about? Is it a distribution that captures how likely a phone or a word piece is used in all of the world's languages?

While the long-tail problem is not properly defined, the class imbalance problem more or less is. There is still a certain amount of ambiguity. For example, what are the classes? Are the classes languages, phones, or word pieces?

Given that the long-tail problem is not defined, it is hard to see why the proposed additions solve the problem. I can understand using a larger language model would help the final performance, but how does this solve the long-tail problem and the class imbalanced problem? The same applies to language adapters. The smoothing technique does have a effect on generalizing to low frequency or even unseen tokens, but the paper does not mention the connection or cite the proper papers.

The paper also ignores the relationships among languages. For example, it is obvious that none of the word pieces in Mandarin are shared with the other languages. It is also the only tonal language. As another example, Tatar is Turkic but uses the Cyrillic script; Turkish is also Turkic but it uses the Latin alphabet; Russian is not Turkic but uses the Cyrillic script. These relationships are important in interpreting the results when training multiple languages together.

Here are a list of detailed comments.

> x \in R^{T,F}

T,F is a rather unconventional notation. I would suggest T \times F.

> KL(y_{ATTN} || y)

Are the y's labels? This is also an unconventional (if not wrong) notation. It should be the the KL of distributions, not labels. Later on, for example in equation (3), y is used as labels.

> equation (3)

\mathcal{Y} is undefined.

> Figure 7 depicts ...

Figure 7 is in the appendix. The main content without the appendix should be as self-contained as possible.

> Let t denote the current time step.

This is confusing. It's actually not the time in the actual speech, but the t-th token.

> A natural adjustment is to scale the raw logits ...

The term logit is misused. Please look it up, stop misusing it, and define the symbols properly.

> equation (6)

The symbol * should really be \times.

> equation (9)

It is confusing to denote the probability as y_t^{adj}. Again, because the bold face y is used as a sequence of labels else where, such as equation (11).

> ... and 2 times gradient accumulation in a single GPU ...

What does this mean exactly? Please elaborate.

> This is due to the human languages share some common sub-phonetic articulatory features (Wang & Sim, 2014) ...

1. This sentence is ungrammatical. 2. This is a well-known fact, and the citation cannot be this recent. 3. No evidence in this paper is shown that this is the actual cause of the improvement. Please state it clearly if this is only a speculation.

> ... even MT models improve the performance of the low-resource languages significantly.

This is not exactly true. For example, the performance on Mandarin actually degrades quite significantly.

> ... compared to the MT, the tail classes ... However, the head classes suffer ...

Are the terms tail classes and head classes defined?

> ... and possibly model overfitting to the tail classes.

This is easy to check. What's the performance on the training set?

> The gains ... of the head languages, although tail languages ...

Again, what are head and tail languages?

---

> ### Author Response · Authors · 2020-11-19
> **Responses to Reviewer 1**
>
> Thank you for the insightful and detailed review. We have thoroughly read your review and have updated our paper to address all reviewer comments. And here, we would answer your questions and concerns.
>
> **Q: Long-tail problem is not properly defined. Is it a distribution that captures how likely a phone or a word piece is used in all of the world's languages?**
> - Thank you for pointing this out. We realized that the problem is not properly defined due to the page limit. We added a paragraph in the introduction (second paragraph) to properly define the problem. We also added a figure (Figure 1) to illustrate the problem further.
> - From the model perspective, the long-tail distribution refers to the skewed subword (word pieces) class distribution of the multilingual data from 11 languages studied in this paper. The skewed distribution is due to two levels of imbalances: the data distribution level and the subword distribution level. First, there are very limited audio samples available on low-resource languages, such as Kyrgyz, Swedish, and Turkish, while the high-resource language data, such as English, French, and Spanish, have vast amounts of data. Second, the distribution of the graphemes or subwords labels follows a long-tailed distribution in ASR since some labels appear significantly more frequent than other labels, even for a monolingual setting. We show in Figure 8 that even with the same amount of training data for each language, the distribution of the subwords of the multilingual token set is still long-tailed. Furthermore, a multilingual system may include languages with various writing scripts other than Latin alphabets, such as Chinese or Cyrillic, that eventually maximize the skewness.
>
> **Q: The smoothing technique does have an effect on generalizing to low frequency or even unseen tokens, but the paper does not mention the connection or cite the proper papers**
> Thanks for the suggestion. We added the statement in our revised paper as suggested and cited the relevant papers under Equation 2.
>
> **Q: I can understand using a larger language model would help the final performance, but how does this solve the long-tail problem and the class imbalanced problem?**
> - The mBERT model and language adapters are employed to enhance the language and acoustic modeling capability for low-resource languages, respectively. The logit adjustment is to explicitly address the class imbalance problem (the distribution of class labels, i.e., multilingual word pieces) regardless of the amount of language resources by adjusting the class distributions. Logit adjustment is complementary to the other two techniques and can be easily applied to other tasks with long-tailed class distributions. We will make this clear in the revised paper.
> - In addition, we don't think a larger language model will help the class imbalance problem. The language itself is inherently long-tailed if you consider all the letters or graphemes. For example, in English, the letter "e" appears much more frequently than the letter "q". Therefore, the resulting word pieces will also manifest such imbalance distributions. This can also be seen in the histogram of Figures 5 and 6. Even with equal numbers of language training data, the multilingual word pieces still have a long-tail distribution, which cannot be fixed by the language model alone.
>
> **Q: The relationships among languages are ignored?**
> Thanks for the suggestion for the relationships of languages. To take your advice and we are currently running more experiments by allowing languages within the same language group to share the same language adapters:
>
> - Grouped by language families
> Romance languages: it fr es
> Germanic languages: en nl sv
> Turkic languages: tr tt ky
> Russian ru
> Chinese zh
>
> - Grouped by written scripts:
> Chinese zh
> Latin: it fr es en nl sv tr
> Cyrillic: ru tt ky
> We will add this study in the revised paper when they are done.

---

> ### Author Response · Authors · 2020-11-19
> **Continued Responses to Reviewer 1**
>
> **Q: What are head and tail languages?**
>
> The head and tail here refer to the amount of training data for multilingual training. The head classes are tokens that have high frequency, otherwise they are classified as tail classes We have divided the languages into high, low, and intermediate resources in Table 1, and all the resource-rich languages can be viewed as the head languages, and the resource-poor languages can be viewed as the tail language.
>
> **Q: Figure 7 is in the appendix. The main content without the appendix should be as self-contained as possible.**
>
> Thanks, we have moved Figure 7 to the main text as Figure 3.
>
> **Q: A natural adjustment is to scale the raw logits … The term logit is misused.**
>
> We define logits as a vector of raw (non-normalized) predictions, and we consistently use it throughout the paper. We remove "raw" from text to avoid any confusion. Also, we have revised Section 2 to remove the confusion caused by using $y$ as both label and distribution.
>
> **Q: Typographical errors**
>
> Thanks for the suggestion, and we have made the changes for the notation of T,F as well as Equation 6. We also modified the definition of $y$ in KL in Section 2 to remove the confusion. We correct the definition of $t$ in Subsection 2.3.
>
> **Q: equation (9) It is confusing to denote the probability as $y_t^{adj}$. Again, because the bold face y is used as a sequence of labels else where, such as equation (11).**
>
> We have revised the paper to use y exclusively for labels.
>
> **Q: Gradient accumulation**
>
> We train our model with a batch size of 32 and accumulate the gradient in two steps to have a larger batch size in a single GPU NVIDIA V100 16GB with Adam optimizer with a warm-up step of 25000.
>
> **Q: This is due to the human languages share some common sub-phonetic articulatory features (Wang & Sim, 2014) ...This sentence is ungrammatical. 2. This is a well-known fact, and the citation cannot be this recent. 3. No evidence in this paper is shown that this is the actual cause of the improvement. Please state it clearly if this is only a speculation.**
>
> We have revised the paper accordingly and cite some previous papers.
>
> **Q: SMT models improve the performance of the low-resource languages significantly. This is not exactly true. For example, the performance on Mandarin actually degrades quite significantly.**
>
> - As for comparing multilingual and monolingual performance, we would like to clarify that the token sets of the monolingual and multilingual are not the same in the original paper. The token set for monolingual is generated only from the specific language and has a much smaller number of target labels. On the other hand, the multilingual training token set is generated from pooled texts from all languages.
> - To make sure, we performed a fair comparison for our multilingual model to the monolingual model after re-training all monolingual models with the same token set as the multilingual setting (5K) and revised the numbers for monolingual in Table 1 to avoid confusion.
>
> **Q: Overfitting check. What's the performance on the training set?**
>
> We performed model decoding on the training data of the tail language "ky".
> The BS model is clearly over-fitted to the tail languages due to up-sampling, for example, the CERs on the training data of "ky" and "sv" are significantly lower than the evaluation data (3.4\% and 4.2\% training vs 13.4\% and 22.8\% evaluation). Compared with BS, A2 also avoids over-fitting to the tail languages, CERs on "ky" and ``"sv" are 8.2\% and 23.6\%, much closer to evaluation CERs

---

### Official Review · AnonReviewer4 · 2020-10-29
**Reasonable approach but unconvincing**

**Rating:** 5
**Confidence:** 4

**Review:**

This paper addresses multi-lingual speech synthesis, where one ASR model is responsible for recognizing speech in multiple languages.  In this example the authors look at 11 languages with between 80 and 4 hours of training data.  The "long-tail problem" (which isn't clearly stated) that this work is addressing is that the discrepancy in available training data leads to a discrepancy in performance.  The paper sets out two goals 1) "to improve the overall performance of multilingual ASR tasks" and 2) (implicitly) to flatten the distribution across languages.

A major challenge in multilingual (or multidomain or multitask) modeling like this is that improvements to the tail often come with degradation at the head.  This work demonstrates this phenomenon clearly.  On the largest languages, English performance degrades from 13.3 to 22.0 and French from 11.5 to 17.7, while on the smallest languages, Kyrghyz improves from 30.0 to 12.1 and Swedish improves from 56.1 to 21.3.  While the language average performance improves from 22.3 (monolingual) to 16.0 (proposed multilingual) it is not at all obvious that there is an application setting where this is clearly preferable.  One way to mitigate this is to pose the problem not as solving universal, multilingual speech recognition, but rather improving performance specifically on tail languages through training on higher resource languages.  If the authors were to focus on improving performance on the 8 languages with 20h or less training data, while including English (en) French (fr) and Spanish (es), but not actually caring whether the high resource languages are improved by multilingual modeling, the results here would be much more compelling.  As written the story is somewhat muddled: On average (where average is taken over language, rather than, say expected usage or the system, or population, etc.) performance improves, but the improvement to lower resource languages comes at the cost of higher resource languages.  Also A2 the proposed system on average does better than standard multilingual training, but only on the 9 lowest resource languages, on English and French A2 actually exacerbates this problem with these higher resource languages showing even larger regressions from monolingual modeling.

Implicit in this approach and task is a desire for the distribution of performance across languages to be more consistent.  I would recommend making this explicit and providing some measure of variance as well as average across languages.  This could be standard deviation (if there is a belief that the performance is normally distributed) or an entropy measure.  But it would provide another dimension over which to optimize when understanding tail performance.

I believe there is a typo or error in Equation 6.  First, there are mismatched subscripts for \pi_y and c_i.  I believe this should be \pi_i or c_y.  Second consider a distribution with three classes and label counts of c = [1, 0, 0], so C=1, n_0 = 2 and N = 3.  Equation 3 would result in \pi = [1/1 - 1/(2*1), 1/1, 1/1] = [1/2, 1, 1] which is not a valid distribution.

Minor comment: Figure 7 is mentioned in Section 2.3 but is only included in the Appendix.  It would be clearer to either describe Figure 7 where it is first mentioned, or present this information in Section 2.3 as forward referring to Appendix material.

---

> ### Author Response · Authors · 2020-11-19
> **Responses to Reviewer 4**
>
> Thank you for the insightful and detailed review. We have thoroughly read your review and have updated our paper to address all reviewer comments. And here, we would answer your questions and concerns.
>
> **Q: One way to mitigate this is to pose the problem not as solving universal, multilingual speech recognition, but rather improving performance specifically on tail languages through training on higher resource languages.**
> - The main motivation of multilingual recognition is to recognize multiple languages with a single model. This not only saves the trouble of creating a separate phone set, language model, and decoder for each language for faster deployment and easier maintenance, the multilingual training will help the individual languages, especially the low-resource languages.
> - As for the comparison of multilingual and monolingual performance, we would like to clarify that the token sets of the monolingual and multilingual are not the same. The token set for monolingual is generated only from the specific language and has a much smaller number of target labels. On the other hand, the multilingual training token set is generated from pooled texts from all languages. Thus, the complexity of the monolingual model is much less than the multilingual model.
> - The monolingual token set has 150 tokens per language, whereas, for multilingual training, there are more than 5K tokens in total (see Table 8). For example, for “en”, there are 243 tokens, and “fr” has 382 tokens.
> - To make sure, we performed a fair comparison for our multilingual model to the monolingual model after re-training all monolingual models with the same token set as the multilingual setting (5K) and revised the numbers for monolingual in Table 1 to avoid confusion. We found that the gap between monolingual and multilingual models for “en” is very small (21.6 vs. 22.0), and for “fr”, A2 improves from 19.8 to 17.7 CER. We reported these numbers in Table 1 of the revised paper. Lastly, the new results demonstrate the advantages of the A2 framework by improving all languages compared to the monolingual models (except for a small degradation of performance for en from 21.6 to 22.0).
>
> **Q: On average performance improves, but the improvement to lower resource languages comes at the cost of higher resource languages. Also A2 the proposed system on average does better than standard multilingual training, but only on the 9 lowest resource languages, on English and French A2 actually exacerbates this problem with these higher resource languages showing even larger regressions from monolingual modeling.**
> - While the best results in the revised Table 1 show some degradation of performance for “en” (21.6 vs. 22.0) with the same multilingual token set, the comparison of SMT and A2 is not fair. SMT is trained with random sampling, and A2s are trained with balanced sampling. A2 should be compared with the balanced sampling version of SMT given in the BS result row.
> - We can clearly see that balanced sampling helps the tail classes and hurts the head classes considerably. With the A2 framework, we can significantly reduce the gap and improve the performance of all languages compared to the multilingual training baseline. Alternatively, one can also compare “SMT” and “SMT+Adjust” performance in Table 2 to appreciate the advantages of A2 in helping improve all languages.
>
> **Q: Typographical error**
> Thank you for pointing this out. After careful examination and checking of our implementation, we revised the Equation 6 to $\frac{C_i}{C} - \frac{1}{(N-n_o)\times C}$, while $c_i > 0$, $\frac{1}{n_o \times C}$, otherwise, in the revised paper.
>
> **Q: Minor comment: Figure 7 is mentioned in Section 2.3 but is only included in the Appendix. It would be clearer to either describe Figure 7 where it is first mentioned, or present this information in Section 2.3 as forward referring to Appendix material.**
> We moved the figure to the main paper. Thank you for your suggestion. We appreciate it.

---

> > ### Comment · AnonReviewer4 · 2020-11-19
> > **Troubling modification of baseline performance**
> >
> > Responses inline.
> > > * The main motivation of multilingual recognition is to recognize multiple languages with a single model. This not only saves the trouble of creating a separate phone set, language model, and decoder for each language for faster deployment and easier maintenance, the multilingual training will help the individual languages, especially the low-resource languages.
> >
> > The initial review did not intend to undermine the motivation of multilingual recognition.  Rather I was suggesting a perspective that would highlight the contributions of this work.  Specifically, it is able to improve performance on lower resourced languages by training with high resource languages.  The approach is less compelling in its ability to recognize higher resourced languages due to the substantial degradation of performance on these.
> >
> > > * As for the comparison of multilingual and monolingual performance, we would like to clarify that the token sets of the monolingual and multilingual are not the same. The token set for monolingual is generated only from the specific language and has a much smaller number of target labels. On the other hand, the multilingual training token set is generated from pooled texts from all languages. Thus, the complexity of the monolingual model is much less than the multilingual model.
> > > * The monolingual token set has 150 tokens per language, whereas, for multilingual training, there are more than 5K tokens in total (see Table 8). For example, for “en”, there are 243 tokens, and “fr” has 382 tokens.
> > > * To make sure, we performed a fair comparison for our multilingual model to the monolingual model after re-training all monolingual models with the same token set as the multilingual setting (5K) and revised the numbers for monolingual in Table 1 to avoid confusion. We found that the gap between monolingual and multilingual models for “en” is very small (21.6 vs. 22.0), and for “fr”, A2 improves from 19.8 to 17.7 CER. We reported these numbers in Table 1 of the revised paper. Lastly, the new results demonstrate the advantages of the A2 framework by improving all languages compared to the monolingual models (except for a small degradation of performance for en from 21.6 to 22.0).
> >
> > I find this decision to be very troubling.  Why should monolingual models be trained to multilingual targets?  The baseline that was included in the original paper -- monolingual acoustic modeling to monolingual sentence piece targets -- was appropriate.  Part of the "cost" of developing a multilingual model is the complexity of needing to recognize multiple languages.  This modified baseline is suggesting that monolingual models (despite being trained only on a single language) should be able to recognize out of language targets, but also must learn that they are out of language.  This is a remarkable requirement for monolingual training.
> >
> > I would like to avoid assigning intention to this modification, and I would hope that it is in the interest of a more transparent understanding of model behavior rather than seeking to present the proposed approach in a more positive light by reporting worse baseline performance.
> >
> > I can't think of any good reason to *remove* the initial monolingual baseline numbers.  If these results -- monolingual modeling to multilingual targets --  are a useful point of comparison between the monolingual baseline and the A2 results, then they should be included *as well*, but not *instead of* the monolingual baseline.  The author's could then attribute the monolingual regressions to the difficulty of making predictions to a larger and more complicated target set.  The original observation that A2 is substantially worse than monolingual training on 'en' and 'fr' should remain.
> >
> > > * While the best results in the revised Table 1 show some degradation of performance for “en” (21.6 vs. 22.0) with the same multilingual token set, the comparison of SMT and A2 is not fair. SMT is trained with random sampling, and A2s are trained with balanced sampling. A2 should be compared with the balanced sampling version of SMT given in the BS result row.
> >
> > I disagree with the claim that the comparison between SMT and A2 is "not fair" because of their training strategy.  While, yes, the balanced sampling exasperates the differences between high and low resourced languages, there is no requirement or expectation for SMT to follow the A2 sampling strategy. It is a fair comparison to say that SMT is a reasonable multilingual baseline, and A2 should be attempting to surpass its performance, not surpass a weakened version of it.  That said, the ablation results in Table 2 are clearly informative to how the Adapt step helps mitigate the impact of balanced sampling, and lead to the conclusion that A2 is able to provide further improvements to lower resourced languages. However, these improvements do come at a modest cost to higher resourced languages (as demonstrated by the comparison to SMT).

---

> > > ### Author Response · Authors · 2020-11-24
> > > **Clarifying the misunderstanding in monolingual settings**
> > >
> > > Thank you again for the feedbacks regarding our revisions. There is probably some misunderstanding in the previous revision and we also found some wrong configurations that caused the huge gap of  monolingual CERs and the detailed comments are given below.
> > >
> > > - The main motivation of multilingual recognition is to recognize multiple languages with a single model. This not only saves the trouble of creating a separate phone set, language model, and decoder for each language for faster deployment and easier maintenance, the multilingual training will help the individual languages, especially the low-resource languages.
> > > > **Your response:**  The initial review did not intend to undermine the motivation of multilingual recognition. Rather I was suggesting a perspective that would highlight the contributions of this work. Specifically, it is able to improve performance on lower resourced languages by training with high resource languages. The approach is less compelling in its ability to recognize higher resourced languages due to the substantial degradation of performance on these.
> > > >> ***Our new response:*** We appreciate your comments and suggestions. We will explain more in the following comments for the gap between the monolingual numbers in the first paper version and A2 systems as we have found that some higher resourced monolingual training in our original version is using a significantly larger training set than the multilingual training.
> > > -As for the comparison of multilingual and monolingual performance, we would like to clarify that the token sets of the monolingual and multilingual are not the same. The token set for monolingual is generated only from the specific language and has a much smaller number of target labels. On the other hand, the multilingual training token set is generated from pooled texts from all languages. Thus, the complexity of the monolingual model is much less than the multilingual model.
> > > -The monolingual token set has 150 tokens per language, whereas, for multilingual training, there are more than 5K tokens in total (see Table 8). For example, for “en,” there are 243 tokens, and “fr” has 382 tokens.
> > > -To make sure, we performed a fair comparison for our multilingual model to the monolingual model after re-training all monolingual models with the same token set as the multilingual setting (5K). We revised the numbers for monolingual in Table 1 to avoid confusion. We found that the gap between monolingual and multilingual models for “en” is very small (21.6 vs. 22.0), and for “fr,” A2 improves from 19.8 to 17.7 CER. We reported these numbers in Table 1 of the revised paper. Lastly, the new results demonstrate the advantages of the A2 framework by improving all languages compared to the monolingual models (except for a small degradation of performance for en from 21.6 to 22.0).
> > > > **Your response:** I find this decision to be very troubling. Why should monolingual models be trained to multilingual targets? The baseline that was included in the original paper -- monolingual acoustic modeling to monolingual sentence piece targets -- was appropriate. Part of the "cost" of developing a multilingual model is the complexity of needing to recognize multiple languages. This modified baseline is suggesting that monolingual models (despite being trained only on a single language) should be able to recognize out of language targets, but also must learn that they are out of language. This is a remarkable requirement for monolingual training.
> > > >> ***Our new response:*** Thanks again for the prompt and insightful comments. There might be some misunderstanding here. By “monolingual models be trained to multilingual targets,” we mean the current monolingual experiments use the subset of tokens from the 5K multilingual targets that belong to the particular language. For example, we used only 243 targets for “en” rather than the whole 5K plus targets. Therefore, the monolingual model is only trained and evaluated on the 243 “en” token set, and there are no “out of language” targets.

---

> > > > ### Author Response · Authors · 2020-11-24
> > > > **Clarifying the misunderstanding in monolingual settings, continued (1)**
> > > >
> > > > > **Your response:** I would like to avoid assigning intention to this modification, and I would hope that it is in the interest of a more transparent understanding of model behavior rather than seeking to present the proposed approach in a more positive light by reporting worse baseline performance.
> > > > I can't think of any good reason to remove the initial monolingual baseline numbers. If these results -- monolingual modeling to multilingual targets -- are a useful point of comparison between the monolingual baseline and the A2 results, then they should be included as well, but not instead of the monolingual baseline. The author's could then attribute the monolingual regressions to the difficulty of making predictions to a larger and more complicated target set. The original observation that A2 is substantially worse than monolingual training on 'en' and 'fr' should remain.
> > > > >> ***Our new response:*** Thank you for your response. Here we would like to clarify some misconfigurations in our monolingual study.
> > > > Firstly, we would like to emphasize that the change is purely for the sake of fair comparison and would like to know how different token sets would impact the model behavior. The purpose of the A2 is not to significantly outperform the monolingual performance for all languages, especially the higher resourced languages; rather, we want to achieve a better-balanced error performance among all languages. We replaced the first version of monolingual results because initially, when we prepared the first response to your comments, we thought the new monolingual setting only differed from the first version in terms of token set. We know how unethical it is to deliberately hide the model limitations with misleading results, and that is certainly not our intention.
> > > >
> > > > >> Having read your new responses and to avoid misunderstanding, we’ve decided to put both sets of results in the table. To explain the gaps of two token sets, we performed another round of inspection of our two sets of monolingual studies. We found a significant factor we overlooked before; in addition to the token set differences, we wrongly reported the initial monolingual baseline results obtained from the monolingual models trained with a much larger data set. These models were initially trained for our initial plan of investigating the unsupervised representation learning for multilingual ASR on the CommonVoice data before we embarked on the A2 framework. For example, “en” was trained on 878 hours (CER 13.3) of data instead of 80 hours (CER 21.6), and “fr” was trained on 273 hours (11.5) of data instead of 40 hours (19.8).
> > > >
> > > > >>For the long-tail problem presented in the paper, we curated a new subset of the original multilingual data with careful partitioning of the train, dev, and test sets for each language. The reason for curating such a subset for our long-tail study is mainly for shorter experiment turnaround time without compromising the long-tail language and word piece class distribution. Since we are working with 11 languages and each multilingual model training takes 2-3 days with a single GPU, but with larger data, we need more than two weeks for each setting with our GPU machines. For now, we would like to clarify this in the revised paper, and the reason why there are significant gaps between the two monolingual settings is because of the training data size. We will produce a new set of monolingual results with the same amount of training data as the multilingual (e.g., 80 hours) but with monolingual tokens rather than the subset of multilingual tokens.  We have kept the best monolingual results trained from the much larger training data together with the training data sizes in the Table 1 for comparison. In addition, we would also conjecture based on the current study that even if the same amount of the larger training data as the best monolingual setting were used for multilingual training, our A2 model could still maintain a similar performance gain over the SMT baseline since the data distribution will be even more imbalance considering the increase in training data size of high resource languages is significantly larger than the lower resource languages (the training data sizes of last three languages are the same for all monolingual settings).

---

> > > > > ### Author Response · Authors · 2020-11-24
> > > > > **Clarifying the misunderstanding in monolingual settings, continued (2)**
> > > > >
> > > > > -While the best results in the revised Table 1 show some degradation of performance for “en” (21.6 vs. 22.0) with the same multilingual token set, the comparison of SMT and A2 is not fair. SMT is trained with random sampling, and A2s are trained with balanced sampling. A2 should be compared with the balanced sampling version of SMT given in the BS result row.
> > > > > >**Your response:** I disagree with the claim that the comparison between SMT and A2 is "not fair" because of their training strategy. While, yes, the balanced sampling exasperates the differences between high and low resourced languages, there is no requirement or expectation for SMT to follow the A2 sampling strategy. It is a fair comparison to say that SMT is a reasonable multilingual baseline, and A2 should be attempting to surpass its performance, not surpass a weakened version of it. That said, the ablation results in Table 2 are clearly informative to how the Adapt step helps mitigate the impact of balanced sampling, and lead to the conclusion that A2 is able to provide further improvements to lower resourced languages. However, these improvements do come at a modest cost to higher resourced languages (as demonstrated by the comparison to SMT).
> > > > > >>***Our new response:*** Thanks for the comment. We agree with your comment that SMT is a decent multilingual baseline. We have revised the paper to keep comparing A2 with it and acknowledge the modest cost for en and fr in the text.

---

### Official Review · AnonReviewer3 · 2020-10-30
**A mix of tricks for an important problem**

**Rating:** 5
**Confidence:** 4

**Review:**

This paper aims to improve multilingual speech recognition on common voice, which contains 18 languages, some of which have little data (which the authors here refer to as the long-tail languages I believe). The problem of multilingual ASR is both a practical one as well as a challenging one from the perspective of multitask learning and fairness, and I'm happy to see work in this area.

The paper proposes 3 techniques that together result in a modest improvement over the baseline on common voice. The 3 include logit re-balancing based on class priors, fusion of a BERT-based language model, and the use of a common and langauge-specific adapter layer in parallel. All of these techniques have been previously explored in slightly different forms for speech problems. They have not been combined in this way before though. To my knowledge, the logit adjustment has not been applied to the long-tail problem in speech recognition.

Pros
- Addresses an important problem in ASR
- Overall, A2 improves over the baseline of balanced sampling by an average of 1% absolute CER, or a relative improvement of 6%. That is a moderate improvement but worthwhile enough to report.
- Introduces class-based logit adjustment to the problem of long tail
- Introduces minor tweaks that lead to improvement, and presents ablation study

Cons
- In large scale models such as this, it is improtant to report the computation requirements of the model in addition the to quality improvements, as often the quality grows with model size. Here there are no comparisons of parameter count here
- Besides the ablation studies, there's not much to be learned on how the changes (dual adapter, logit adjustment, or the way mbert is fused) helped the quality. It would be nice to report a few failed versions that the authors tried to learn more about what works and what doesn't.
- Overall the changes do not improve significantly over the baseline. Also there should be more competing baselines to consider, other than the adapter layers of Kannan et al. There's the multi-headed decoder approach of Pratap et al. or the language ID injection approach of Li et al. "Multi-Dialect Speech Recognition with a Single Sequence-to-Sequence Model".
- It's quite unclear what the long tail refers to in this paper. Does it refer to the languages that have little data? Or does it refer to words that are rare or often misclassified? Most of the paper leads me to believe in the former, but Figures 5 and 6 in the appendix lead me to believe in the latter since the histograms are so dense.
- There's a lack of specific examples that illustrate how the incorporation of the various techniques in this paper show an improvement in the transcription. Showing specific transcriptions would be convincing in terms how showing the wins from these techniques...

Other comments:

What is meant by the fourth bullet point in the contributions? Is there a new dataset? I do not understand the contribution

The use of previous tokens as input, i.e. not using teacher forcing, during the later stages of training (Eq. 10) is unconventional. It would be more convincing if the author discussed this a little more, including why it improves quality.

It's unclear how x_{CTC} is defined in fig 1. Is it the output of the encoder?

Likewise it's unclear how the function f is defined in fig 1. Is it the same function and weights (assuming a linear transformation from the previous layer) for f(x_CTC) and f(y'_ATTN, h_enc)?

Fig 7 and comments to it should be moved to the main paper. It is essential for understanding of how mbert is integrated into the decoder as that is a big part of the contribution.

The grammar throughout the document is occasionally off which distracts from the content. Needs polish.

---

> ### Author Response · Authors · 2020-11-19
> **Responses to Reviewer 3**
>
> Thank you for the insightful and detailed review. We would answer your questions and concerns as follows:
>
> **Q: In large scale models such as this, it is important to report the computation requirements of the model in addition to quality improvements, as often the quality grows with model size.**
>
> We have added the parameters count in the Ablation Studies section (Tables 3, 4, and 5).
>
> **Q: Besides the ablation studies, there's not much to be learned on how the changes helped the quality.**
>
> - For logit adjustment, we showed several tau values for performance comparison in a new subsection 3.3.4. Also, we found that the training or inference phase logit adjustment cannot be applied together, which will break the probability distribution and yield significantly worse performance.
> - Our initial attempt of mBERT with freezing self-attention (only cross-attentions are updated) yields even worse performance than the baseline random initialized decoder weights (We will put this observation in our revised paper). We have also replaced the mBERT with a more advanced XLM-R pretrained language model to study the impacts of the pretrained language models on the proposed A2 framework.
> - We have added more experiments with language group adapters, where the languages of the same language group (e.g., “es” “it” “fr” of the Romance language) share the same adapter parameters to study whether a more robust language adapter can be obtained with more training data and whether it can benefit each language of the group. The experiments and discussions are presented in Table 4.
>
> **Q: Also there should be more competing baselines to consider, other than the adapter layers of Kannan et al.**
>
> We built the language ID injection (LID) system from Li et al for comparison. We show the results in Table 1. We found that our A2 model outperforms LID (16.0 vs. 17.1 CER), and if word piece class imbalance adjustment is applied, a significant CER improvement is achieved (16.6 CER).
>
> **Q: It's quite unclear what the long tail refers to in this paper. Does it refer to the languages that have little data? Or does it refer to words that are rare or often misclassified? Most of the paper leads me to believe in the former, but figures the appendix leads me to believe in the latter since the histograms are so dense.**
>
> From the model perspective, the long-tail distribution refers to the skewed sentence piece classes of the multilingual data. The skewed distribution stems from two levels of imbalances: the language training data size (number of training samples) and the sentence piece distribution (number of tokens) level. To better solve the long-tail problem, we need to 1) model the low resource languages robustly 2) address the sentence piece class long-tail distribution properly.
> We show in Figure 8 that even with the same amount of training data for each language, the distribution of the subwords of the multilingual token set is still long-tailed. We have made this clear in the introduction section of the revised paper. In addition, we have adjusted the histogram plots to make them less dense.
>
> **Q: There's a lack of specific examples that illustrate how the incorporation of the various techniques in this paper show an improvement in the transcription. Showing specific transcriptions would be convincing in terms how showing the wins from these techniques.**
>
> We added examples in the Appendix E in the revised paper.

---

> ### Author Response · Authors · 2020-11-25
> **Continued Responses to Reviewer 3**
>
> **Q: What is meant by the fourth bullet point in the contributions? Is there a new dataset? I do not understand the contribution**
>
> To the best of our knowledge, there is no existing benchmark for ASR with a focus on long-tailed class distribution. CommonVoice is a public dataset with multiple languages. However, there is no standard partition of data for the long-tail distribution study. Therefore, our contribution lies in curating a subset of CommonVoice data as a benchmark for multilingual speech recognition. This work will help future researchers have a standard partition for benchmarking their multilingual ASR systems.
>
> **Q: The use of previous tokens as input, i.e. not using teacher forcing, during the later stages of training (Eq. 10) is unconventional. It would be more convincing if the author discussed this a little more, including why it improves quality.**
>
> Scheduled sampling is widely used in training end-to-end ASR systems to reduce the mismatch of training and inference. During inference, the prediction of the current token depends on the previous ones due to the autoregressive decoder. During training, we can use teacher forcing to use the ground-truth label of the previous token for forward and loss computation. During inference, there is no label for the previous token, and we can only rely on the prediction and beam search to decode the current token. To reduce such mismatches, during later training stages, at a certain probability (e.g., 0.3), instead of using the ground-truth label, we use the previous token with the largest posterior to let the model handle such discrepancies, which is helpful for the final decoding.
>
> **Q: It's unclear how x_{CTC} is defined in fig 1. Is it the output of the encoder? Likewise it's unclear how the function f is defined in fig 1.**
>
> Yes, x_{CTC} is the output. Function f is the linear function for computing the logits before applying the softmax. We have updated the paper to make it more clear.
>
> **Q: Fig 7 and comments to it should be moved to the main paper. It is essential for understanding how mbert is integrated into the decoder as that is a big part of the contribution.**
>
> We moved Fig 7 to the main paper as Figure 3.

---

### Official Review · AnonReviewer5 · 2020-11-05
**Simple method with comprehensive experiments**

**Rating:** 6
**Confidence:** 3

**Review:**

This paper studies multilingual ASR with a focus on the long tail problem. A new method using dual adapters is proposed. Although there are several ingredients of the method, their effectiveness are all verified in detailed ablation studies. Therefore, I believe the results shown in this paper are valuable for future work.

Pro:
1. The structure of dual adapters is novel.
2. To the best of my knowledge, this is the first work to verify the effectiveness of pretrained models in multilingual ASR.
3. The paper contains detailed experiments.

Con:
1. The framework combines many techniques together and it is hard to tell if any one of those is the 'silver bullet'.
2. Some design/hyperparameter choices are rather magical.

Questions:
1. Why did you choose to use distill-mBERT over other alternatives (mBERT, XLM etc.)? Would you expect more gain if using a larger model such as XLM-R?
2. Recent work [1] shows negative interference can impact low-resource languages in multilingual models. However, it seems like the opposite is true here: multilingual models can improve even high-resource languages (e.g. IT). Do you have any idea why?


[1] On negative interference in multilingual models: findings and a meta-learning treatment. Wang et al., EMNLP 2020.

---

> ### Author Response · Authors · 2020-11-19
> **Responses to Reviewer 5**
>
> Thank you for the comments. We have updated our paper to address your comments. We would answer your questions and concerns as follows:
> **Q: The framework combines many techniques together and it is hard to tell if any one of those is the 'silver bullet'.**
> Our three technical contributions include:
> 1. use of a pre-trained multilingual language model,
> 2. dual adapters, and
> 3. logit adjustments were used to improve the multilingual ASR.
> Our studies showed that all three approaches complement each other. The first two techniques are employed to enhance the language and acoustic modeling capability for low-resource languages. Comparing these two techniques, we found that dual adapters are more effective than pre-trained language models (BS + mBERT vs. BS + Dual-Adapters). The third technique, the logit adjustment, addresses the class imbalance problem (the distribution of class labels, i.e., multilingual word pieces) regardless of the amount of language resources. Logit adjustment is complementary to the other two techniques and can be easily applied to other tasks with long-tailed class distributions. To sum up, dual adapters and logit adjustments are the two most important techniques for the success of the A2 framework, while the improvement of the mBERT language model comes at the cost of larger models and heavier computations.
> We have discussed the effectiveness of each technical contribution in our Ablation Studies section.
>
> **Q: Some design/hyperparameter choices are rather magical.**
> We use a fixed training hyperparameter set in our speech recognition model to have a fair comparison. We also added a new subsection to describe better how we find the best hyper-parameter $\tau$ for our class imbalance adjustment.
>
> **Q: Why did you choose to use distill-mBERT over other alternatives (mBERT, XLM etc.)? Would you expect more gain if using a larger model such as XLM-R?**
> Distilled-mBERT was chosen for its smaller size and decent multilingual language modeling performance to speed up the experiments since we are dealing with 11 languages. Our studies showed that the improvement of distill-mBERT (BS+Dual-Adapters and BS+Dual-Adapters+mBERT in Table 2) is only achieved if better acoustic models are used. Although a pre-trained language model has a better language generation capability, for speech recognition applications, aligning the acoustic and text space is also crucial.  Considering the improvement of the distill-mBERT is not as significant as the other two techniques, we think with a larger model like mBERT, XLM-R, the performance gain over the current system with distilled-mBERT will not be significant due to the two considerations:
> with increased model parameters, it will be even more demanding in aligning the text and acoustic space, there will be a vast increase in computation (4x for XLM-R base) for training, and the decoding speed will also suffer significantly.
>
> We have trained a model with an XLM-R decoder and compared it with the mBERT results. Our conjectures were verified in Table 2 as XLM-R does not provide ASR performance gain compared to distilled-mBERT.
>
> **Q: Negative interference can impact low-resource languages in multilingual models. However, it seems like the opposite is true here: multilingual models can improve even high-resource languages (e.g. IT). Do you have any idea why?**
> For speech recognition, different languages share similar articulatory features (sub-phonetic) on how a phoneme is produced, for example, place of articulation, production manner, etc. Multilingual training encourages the model to learn such low-level features, which may be beneficial to all languages, including the resource-rich languages. In addition, languages within the same language group, e.g., “it” (Italian) and “es” (Spanish), share even more similarities in terms of vocabulary, pronunciation, and grammar. The additional data from the same language group can further improve the performance of a language. For “it”, 20 hours of data is not good enough for a decent ASR system, and we categorized it as an intermediate language. Our results in Table 1 show that our A2 system can improve all languages compared to the baseline multilingual training with balanced sampling, thanks to the logit adjustment.

---

### Author Response · Authors · 2020-11-19
**General Response to All Reviewers**

We have uploaded a new version of our paper to address reviewer concerns. We thank all reviewers for the comments and feedback to improve our paper. Here are the summary of the changes:
- Revised the paper considerably in terms of writing, including more precise word usages, fixing grammar errors, fixing typos
     - Clarified the long-tail problem and added a figure; see Figure 1 and the introduction's second paragraph.
     - Corrected Eq 6.
     - Added description of smoothing based on the reviews’ suggestions.
     - Moved mBERT picture to the main paper as suggested by many reviewers
     - Added model parameters in the experimental result tables
     - Added transcription examples and analysis in Appendix E.
- Add more baselines and experiments to make the study more comprehensive.
     - Added a new baseline as suggested by R3, language conditioning with one-hot language vectors (LID) from Li, et al. [1], see Table 1
     - Added a new ablation subsection to show the effect of $\tau$ in the imbalance class adjustment in Appendix D
     - Retrained monolingual models with the same vocabulary as multilingual models for a fair comparison. We also presented the training data sizes in Table 1 to explain the gaps in the CER performance of A2 compared to the best monolingual systems, which are trained on a much larger dataset.
     - Added a set of new adapters for language groups suggested by R1 by allowing languages within the same language group to share the same language adapters. Detailed experiments results and analysis of language groups are in the ablation study 3.3.2
     - Showed A2 can avoid the model overfitting to the tail languages under the balanced sampling.
- A more advanced pretrained language model XLM-R suggested by R5 is used in place of the distilled-mBERT to study the impacts of pretrained languages; see Table 2.

[1] Kannan, et al. Large-scale multilingual speech recognition with a streaming end-to-end model. Interspeech.

---

### Decision · Program_Chairs · 2021-01-07
**Final Decision**

**Decision:**

Reject

**Comment:**

As one of the reviewers' comment, the paper presents "a mixed of tricks" for the multilingual speech recognition, which includes 1) the use of a pretrained mBERT, 2) dual-adapter and 3) prior adjusting.
First, the relative gains of the pretrained mBERT is marginal (Section 3.3.1). Secondly, using 1) on top of 2) is unnecessary.
These confuses the reader about what the conclusion of the paper is.
It would be better if choosing one aspect of the problem and investigate it deeper.

The decision is mainly because of the lack of novelty and clarity.